∂ | **Open Peer Review** | Microbial Ecology | Research Article

# Species-specific ribosomal RNA-FISH identifies interspecies cellular-material exchange, active-cell population dynamics and cellular localization of translation machinery in clostridial cultures and co-cultures

**John D. Hill,**[1] **Eleftherios T. Papoutsakis**[1]

**ABSTRACT**  The development of synthetic microbial consortia in recent years has revealed that complex interspecies interactions, notably the exchange of cytoplasmic material, exist even among organisms that originate from different ecological niches. Although morphogenetic characteristics, viable RNA and protein dyes, and fluorescent reporter proteins have played an essential role in exploring such interactions, we hypothesized that ribosomal RNA-fluorescence *in situ* hybridization (rRNA-FISH) could be adapted and applied to further investigate interactions in synthetic or semisynthetic consortia. Despite its maturity, several challenges exist in using rRNA-FISH as a tool to quantify individual species population dynamics and interspecies interactions using high-throughput instrumentation such as flow cytometry. In this work, we resolve such challenges and apply rRNA-FISH to double and triple co-cultures of *Clostridium acetobutylicum, Clostridium ljungdahlii,* and *Clostridium kluyveri*. In pursuing our goal to capture each organism's population dynamics, we demonstrate dynamic rRNA, and thus ribosome, exchange between the three species leading to the formation of hybrid cells. We also characterize the localization patterns of the translation machinery in the three species, identifying distinct, dynamic localization patterns among them. Our data also support the use of rRNA-FISH to assess the culture's health and expansion potential, and, here again, our data find surprising differences among the three species examined. Taken together, our study argues for rRNA-FISH as a valuable and accessible tool for quantitative exploration of interspecies interactions, especially in organisms which cannot be genetically engineered or in consortia where selective pressures to maintain recombinant species cannot be used.

**IMPORTANCE**  Though dyes and fluorescent reporter proteins have played an essential role in identifying microbial species in co-cultures, we hypothesized that ribosomal RNA-fluorescence *in situ* hybridization (rRNA-FISH) could be adapted and applied to quantitatively probe complex interactions between organisms in synthetic consortia. Despite its maturity, several challenges existed before rRNA-FISH could be used to study *Clostridium* co-cultures of interest. First, species-specific probes for *Clostridium acetobutylicum* and *Clostridium ljungdahlii* had not been developed. Second, "state-of-the-art" labeling protocols were tedious and often resulted in sample loss. Third, it was unclear if FISH was compatible with existing fluorescent reporter proteins. We resolved these key challenges and applied the technique to co-cultures of *C. acetobutylicum*, *C. ljungdahlii*, and *Clostridium kluyveri*. We demonstrate that rRNA-FISH is capable of identifying rRNA/ribosome exchange between the three organisms and characterized rRNA localization patterns in each. In combination with flow cytometry, rRNA-FISH can capture sub-population dynamics in co-cultures.

Address correspondence to Eleftherios T. Papoutsakis, epaps@udel.edu.

The authors declare no conflict of interest.

See the funding table on p. 21.

10.1128/msystems.00572-24  **1**

KEYWORDS *Clostridium ljungdahlii*, *Clostridium acetobutylicum*, *Clostridium kluyveri*, rRNA-fluorescence *in situ* hybridization, flow cytometry, subcellular localization, cytoplasmic exchange, heterologous, interspecies cell fusion

Industrial microbiology has largely employed axenic culture strategies with selected wild-type or genetically engineered organisms or complex, naturally occurring microbial consortia, such as is the case in the dairy industry. For example, *Clostridium acetobutylicum* was grown historically in pure cultures to produce acetone and butanol from molasses before inexpensive petrochemicals made the process unprofitable in the mid-1900s. The chemical industry is returning to fermentation as a way to produce biofuels, chemicals, and bio-hydrogen more ecologically (1). In contrast to traditional approaches, synthetic microbial consortia, that is, consortia consisting of two or more organisms that are not necessarily naturally co-existing, is a promising alternative (2, 3). Synthetic consortia offer some benefits over pure cultures such as the division of labor, modularity, and the compartmentalization of incompatible metabolic reactions (2, 3). Using clostridial organisms, the co-culture approach for production of chemicals has attracted interest during the last few years. Industrially interesting Clostridia are categorized in groups based on their general substrate and metabolic characteristics (4). Cellulolytic species can directly utilize lignocellulosics. Solventogens (an ill-defined term), such as *Clostridium acetobutylicum*, ferment carbohydrates into solvent molecules. Acetogens, such as *Clostridium ljungdahlii*, can fix $CO_2$ via the Wood-Ljungdahl pathway (WLP). Chain elongators, such as *Clostridium kluyveri*, convert short primary alcohols and carboxylic acids into longer-chain fatty acids. By combining species from different groups, the co-culture can be adapted to a variety of feedstocks, including syngas and lignocellulosic biomass, and produce desirable commodity chemicals and fuel molecules (1). In many cases, clostridial organisms operate synergistically allowing for the production of novel metabolites with more efficient substrate conversion (5, 6).

Synthetic consortia provide a uniquely controlled environment to observe interspecies relations which may have been obscured by the inherent complexity of naturally occurring consortia. A variety of novel interspecies interactions have been revealed, including the formation of nanotube bridges, and cytoplasmic exchange mediated by cell fusion (3). Among these, cytoplasmic exchange is arguably the least expected. Benomar et al. first demonstrated that exchange of proteins between *Desulfovibrio vulgaris* and *C. acetobutylicum* can take place under nutritional stress conditions and is mediated by cell-to-cell contact (7). Based on scanning electron micrographs, the cells remain physiologically distinct from one another during these events (7). Subsequent work emphasized the role of energetic coupling (i.e., contact mediated exchange of essential metabolites) between the two organisms and implicated the quorum sensing molecule autoinducer-II as the molecular basis of the interaction (8). Using anaerobic fluorescent reporter proteins and protein dyes, our lab demonstrated heterologous cell fusion-driven cytoplasmic exchange between *C. acetobutylicum* and *C. ljungdahlii* (9). Unlike Benomar et al.'s system, *C. acetobutylicum* and *C. ljungdahlii* contact each other pole-to-pole, and that interaction can lead to heterologous, interspecies cell fusion and the formation of hybrid cells, cells that "contain uniformly distributed proteins and RNA from both organisms" (9). Most recently, it was shown that the cellular material exchange includes plasmid and chromosomal DNA between *C. acetobutylicum* and *C. ljungdahlii* at frequencies comparable to transduction and conjugation in Gram-positive organisms (10). PacBio sequencing revealed that, in some instances, *C. acetobutylicum* incorporated *C. ljungdahlii* plasmid and chromosomal DNA into its own chromosome (10). The mechanistic nature of the interaction and its implications on culture stability, especially on the long-term genetic stability, remain unexplored.

Does cytoplasmic exchange and interspecies cell fusion occur among other pairs of organisms? Of interest to our work are the interactions of *C. acetobutylicum* and *C. ljungdahlii* with *C. kluyveri*, the model organism which carries out chain elongation of linear carboxylic acids up to $C_8$. Work characterizing the metabolic consequences of *C.*

*acetobutylicum*-*C. kluyveri* and *C. ljungdahlii*-*C. kluyveri* co-cultures has been published largely focusing on the metabolic potential of the co-cultures (6, 11). Our 2022 paper provides the first evidence for cytoplasmic material exchange between *C. acetobutylicum* and *C. kluyveri* (6). We sought to explore the interspecies relationships, especially cytoplasmic exchange, between these organisms. However, the transformation of *C. kluyveri* has yet to be reported in the literature. It cannot be made to express fluorescent proteins which has been essential for cytoplasmic tracking in the aforementioned studies. We hypothesized that ribosomal RNA-fluorescence *in situ* hybridization (rRNA-FISH) could be used as an alternative marker for tracking rRNA exchange enabled by heterologous cell fusion. An rRNA-FISH probe is comprised of a short DNA sequence (16 bp–40 bp) which bears homology to a target rRNA sequence. A fluorescent molecule, such as the Cy or Alexa Fluor family of fluorophores, is attached to the 5´end of the DNA probe. During hybridization, the probe forms a duplex with the target rRNA molecule, thus marking it fluorescently. rRNA-FISH has been applied to environmental microbiology since the 1990s, where it is used to identify the presence of different taxonomic groups in naturally occurring consortia (12–14). The DNA probes target sequences of the rRNA that are both unique to and ubiquitous among the targeted taxonomic groups (15). Species-specific rRNA-FISH is an embodiment of rRNA-FISH wherein the FISH probes bear homology to a species-specific region of the rRNA. Thus, individual species of bacteria can be identified among closely related organisms. Recently, it was applied to synthetic consortia to track the subpopulation dynamics of a co-culture containing *Clostridium carboxidivorans* (an acetogen) and *C. kluyveri* (16). Another important facet of rRNA-FISH in the context of synthetic consortia engineering is its unexplored potential to be used for assessing the overall culture health. Several studies have explored the relationship between cell viability and rRNA-FISH labeling, though, in general, cell viability is difficult to determine for bacteria, especially in Gram-positive organisms (17). In *Escherichia coli*, rRNA-FISH was able to determine cell viability as well as other typical methods such as the BacLight Live/Dead and 5-Cyano-2,3-ditolyl tetrazolium chloride assays when cells were exposed to UV light or heat, though all three methods overestimated the number of viable cells (18). In other words, too, rRNA-FISH has been assessed to overestimate the percentage of viable cells in a sample (18–20). In other words, rRNA-FISH tends to give false-positives, rather than false-negatives. Therefore, it cannot be used as a viability assay *per se*, though cell viability is a poorly defined parameter in general. However, unlabeled cells are almost certainly non-viable as lacking a high concentration of rRNA (virtually all of which is bound in ribosomes, as we discuss below). We will examine this issue here using our data.

Here, we designed and validated novel species-specific rRNA-FISH probes for *C. acetobutylicum* and *C. ljungdahlii*. These probes target the 23S rRNA, the primary scaffold rRNA of the large ribosomal subunit. As the "state-of-the-art" hybridization procedure is complex and may lead to cell loss during processing, we aimed to simplify the protocol, removing in total 12 centrifugation and resuspension steps without any apparent loss of labeling efficacy. We validated that multiplexing with a fluorescent reporter protein, HaloTag, and CellTracker protein dyes, is possible under the new protocol. In addition to tracking the individual species populations in clostridial co-cultures, we show that rRNA-FISH can assess a culture's growth ability (or "health"). We demonstrate the potential of rRNA-FISH for identifying hybrid cells (i.e., cells containing rRNA from two different species) in a co-culture of *C. ljungdahlii* and *C. kluyveri* and a triple co-culture with *C. acetobutylicum,* thus enlarging the number of microbial pairs in which interspecies exchange of cellular material takes place. Finally, we report the first rRNA/ribosome cellular localization studies in *C. acetobutylicum*, *C. ljungdahlii*, and *C. kluyveri*.

## RESULTS

### Validation of probe specificity and optimization of an in-solution hybridization protocol

A species-specific rRNA-FISH probe set for *C. kluyveri,* ClosKluy, has previously been reported, but no suitable probes exist for *C. acetobutylicum* or *C. ljungdahlii* (21). Two sets of rRNA-FISH probes, ClosAcet and ClosLjun, were developed to target a species-specific region within the 23S rRNA of *C. acetobutylicum* and *C. ljungdahlii*, respectively. We used an approach similar to Fuchs et al. (22) to determine suitable hybridization conditions for ClosLjun, ClosAcet, and ClosKluy. Formamide disrupts the hydrogen bonding which holds the DNA-RNA duplex together. We tracked the median fluorescence of labeled *C. ljungdahlii, C. acetobutylicum*, and *C. kluyveri* at increasing formamide concentrations to determine the maximum stringency (i.e., maximum formamide concentration) that allowed strong labeling (Fig. 1A through C). For instance, *C. acetobutylicum* was brightly labeled using ClosAcet at or below a formamide concentration of 20% (Fig. 1B). At higher formamide concentrations, the cells became less bright, suggesting that the increasing stringency was preventing ClosAcet from labeling *C. acetobutylicum* rRNA (Fig. 1B). *C. ljungdahlii* and *C. kluyveri* were separately hybridized with ClosAcet under the same conditions to demonstrate the absence of off-target labeling when using ClosAcet with these organisms (Fig. 1B). Analogous experiments were performed for the ClosLjun (Fig. 1A) and the ClosKluy probe sets (Fig. 1C). All three sets of probes demonstrated sufficient specificity and brightness at a formamide concentration of 20%.

Further experimentation was necessary to ensure, first, the selectivity of the probes against off-target labeling, second, a high signal-to-noise ratio of the probes against autofluorescence, and third, the absence of bleed-over of the probe's fluorescent signal into neighboring channels. Three biological replicates of each species were grown, sampled during exponential-phase growth, fixed, and incubated with ClosAcet, ClosLjun, and ClosKluy simultaneously under optimized hybridization conditions (20% formamide; discussed in the next section). These samples were compared to unlabeled negative controls to quantify autofluorescence. For all species, hybridization produced a strong shift in population-level fluorescence when analyzed on the fluorescent channel corresponding to that species' unique probe. For instance, labeled *C. acetobutylicum* (*Cac*) exhibited strong fluorescence on the "ClosAcet Channel" when compared to the unlabeled, negative control, indicating that autofluorescence is negligible (Fig. 1D.i). No strong shift was observed for incompatible species/channel combinations, indicating the absence of bleed-over or off-target labeling. For instance, the fluorescence of labeled *C. acetobutylicum* population overlapped with the negative control population when analyzed on the "ClosKluy Channel" (Fig. 1D.ii) and the "ClosLjun Channel" (Fig. 1D.iii) despite having been incubated with all three probe sets during hybridization. Further experimentation demonstrated the specificity of the probes against *C. ljungdahlii* (Fig. 1E.i, ii, and iii) and *C. kluyveri* (Fig. 1F.i, ii, and iii). Quantitative population data are presented in Tables S1 and S2 for the labeled and unlabeled mono-cultures, respectively. Analogous confocal microscopy tests were performed on the same samples to validate the results obtained by flow cytometry (Fig. S1 to S3). Fluorescence was only observed between corresponding species/channel combinations, but no fluorescent signal could be detected for unlabeled cells and incompatible species/channel combinations. Therefore, fluorescent signals appearing on microscopic images represent the presence of the targeted rRNA.

rRNA-FISH can be performed on slides or in-solution. In-solution rRNA-FISH is attractive because the sample can be analyzed using flow cytometry. When implemented this way, species-specific rRNA-FISH can be used to track subpopulation dynamics in co-culture (21). The major drawback to in-solution rRNA-FISH is that each treatment step requires centrifugation and resuspension which is time consuming and risks the loss of sample from incomplete centrifugation. Additionally, repeated centrifugation and incomplete resuspension could cause cell aggregation in the sample. The state-of-the-art

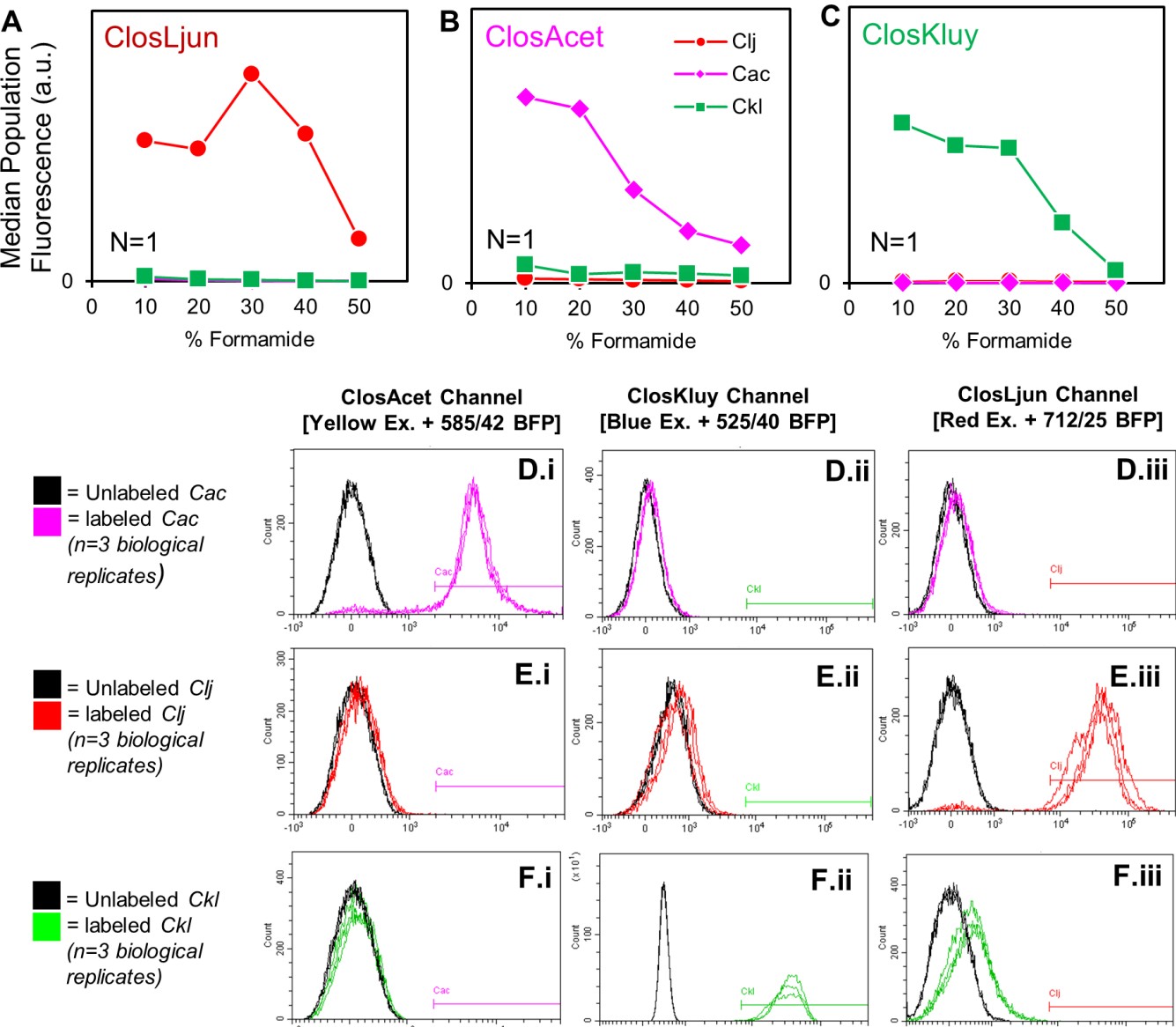

FIG 1 Optimization of formamide concentration and probe specificity. (A–C) *C. acetobutylicum* (Cac), *C. ljungdahlii* (Clj), and *C. kluyveri* (Ckl) were hybridized with each probe at formamide concentrations between 10% and 50% to determine ideal stringency. The median population fluorescence intensity (in arbitrary units, a.u.) was plotted for each sample on a linear axis indexed at 0. (A) ClosLjun selectively binds to *C. ljungdahlii*'s rRNA between 10 and 40% formamide, with optimal fluorescence at 30% formamide. (B) ClosAcet selectively binds to *C. acetobutylicum*'s rRNA between 10% and 20% formamide, with optimal fluorescence at 20% formamide. (C) ClosKluy selectively binds to *C. kluyveri*'s rRNA between 10% and 30% formamide, with optimal fluorescence at 10% formamide. (D) *C. acetobutylicum* was hybridized with ClosAcet, ClosKluy, and ClosLjun simultaneously at 20% formamide concentration and interrogated via flow cytometry on the channels corresponding to each probe's fluorescent marker. The *x*-axis is fluorescent intensity, and the *y*-axis is the number of events at that intensity. "Labeled" samples were compared to "unlabeled" samples which underwent the same hybridization procedure but without any probes. Panel E shows anexperiment analogous to panel D performed in *C. ljungdahlii*. Panel F shows an experiment analogous to panel D performed in *C. kluyveri*. For each species, only the corresponding probe induced a shift in the population's fluorescence. The gating strategy established from this set of experiments was maintained throughout the work.

approach for in-solution rRNA-FISH among Clostridia includes a fixation step (2–4 centrifugation/resuspension [C/R] steps), followed by dehydration (4 C/R steps), lysozyme treatment (2 C/R steps), further dehydration (4 C/R steps), hybridization, and washing steps (4 C/R steps) (16). During preliminary experiments, we discovered that many of the "canonical" prehybridization steps for Gram-positive organisms are either not necessary or even detrimental in some cases. Fixation with paraformaldehyde (PFA)

did not improve fluorescent signal compared to ethanol fixation, which is simpler and avoids the use of PFA, a known health hazard. Lysozyme treatment did not significantly improve signal in PFA-fixed *C. acetobutylicum* and actually led to a complete loss of signal when fixed with ethanol (data not shown). In Bäumler et al.'s work, a dehydration step was included prior to hybridization as this was thought to better draw the probe in via osmotic forces (16, 23). We found that omitting the dehydration step prior to hybridization had no appreciable effect on any of the three organisms (Fig. S4A). Therefore, all dehydration and the lysozyme permeabilization steps were omitted from our protocol, saving 12 C/R steps in total.

Hybridization conditions were screened to optimize the strength and speed of the probe binding in a manner similar to Wallner et al. using 20% formamide (24). We measured the time to maximum fluorescence for three probe concentrations: 0.1 mM (~0.61 ng/mL), 0.5 mM (3.1 ng/mL), and 1 mM (6.1 ng/mL). For each probe concentration and time point, cells from three biological replicates were sampled. The average of the three median population fluorescent intensity is plotted against time in Fig. S4B and C for *C. acetobutylicum* and *C. ljungdahlii,* respectively. Maximum fluorescence was reached between 3 and 5 h, which is consistent with other reports of rRNA-FISH in Firmicutes (25). We therefore conclude that the omission of prehybridization steps likely had a negligible effect on the rate of probe binding. Five hours was sufficient to obtain a high fluorescent signal for both species with a probe concentration of 1 mM. Kinetic studies for *C. kluyveri* were omitted on the basis that previous work had reported success with 5 h incubations (21).

## In-solution rRNA-FISH is compatible with anaerobic fluorescent reporter HaloTag and CellTracker Deep Red protein dye

Previously our lab has adapted the HaloTag fluorescent reporter protein system for use in Clostridia, since other, more conventional reporters (e.g., green fluorescent protein [GFP], mCherry, and flavin-binding proteins) have been used with little or no success as they require oxygen and/or have weak fluorescence (26). HaloTag is a small protein (33 kDa) which is not fluorescent on its own but forms covalent bonds with a wide variety of fluorescent ligands (27). We verified that our newly developed rRNA-FISH labeling protocol is compatible with HaloTag in HaloTag expressing strains: *C. acetobutylicum*-p100ptaHalo and *C. ljungdahlii*-p100ptaHalo (26). Cells from exponentially growing cultures were incubated with their respective "no-wash" HaloTag ligand prior to fixing and rRNA-FISH labelling. For *C. acetobutylicum*-p100ptaHalo, Janelia 646, a far red-emitting ligand, was used since its fluorescent signal does not overlap with Cy3, the yellow-emitting fluorophore of the ClosAcet probe set. For *C. ljungdahlii*-p100ptaHalo, Janelia 549, a yellow-emitting ligand, was used since its fluorescent signal does not overlap with Cy5.5, the far red-emitting fluorophore of the ClosLjun probe set. Figure 2A and B shows the overlapping signal from both cells. HaloTag expression requires genetic modification which is not available in *C. kluyveri*. CellTracker Deep Red (Deep Red) has been used by our lab to follow intercellular protein exchange (9). Deep Red forms covalent bonds with amine residues of proteins through a succinimidyl ester reactive group. To test if this, too, would survive in-solution rRNA-FISH, we labeled *C. kluyveri* with Deep Red prior to hybridization which resulted in strong double-labeled *C. kluyveri* cells as seen in Fig. 3C. Together, we conclude that our in-solution rRNA-FISH protocol is compatible with other fluorescent labeling techniques which rely on covalent bonds.

## The *C. acetobutylicum* time course of rRNA-FISH illustrates severe attenuation of translation in stationary phase due to commitment to sporulation and loss of growth potential

Previous work had determined that during batch cultivation of *C. acetobutylicum,* the number of colony-forming units (CFU) per unit culture volume decreases sharply once the culture reaches stationary phase (28). Since the $OD_{600}$ and cell density remain constant during stationary phase, the decrease in CFU/mL has two potential sources: one

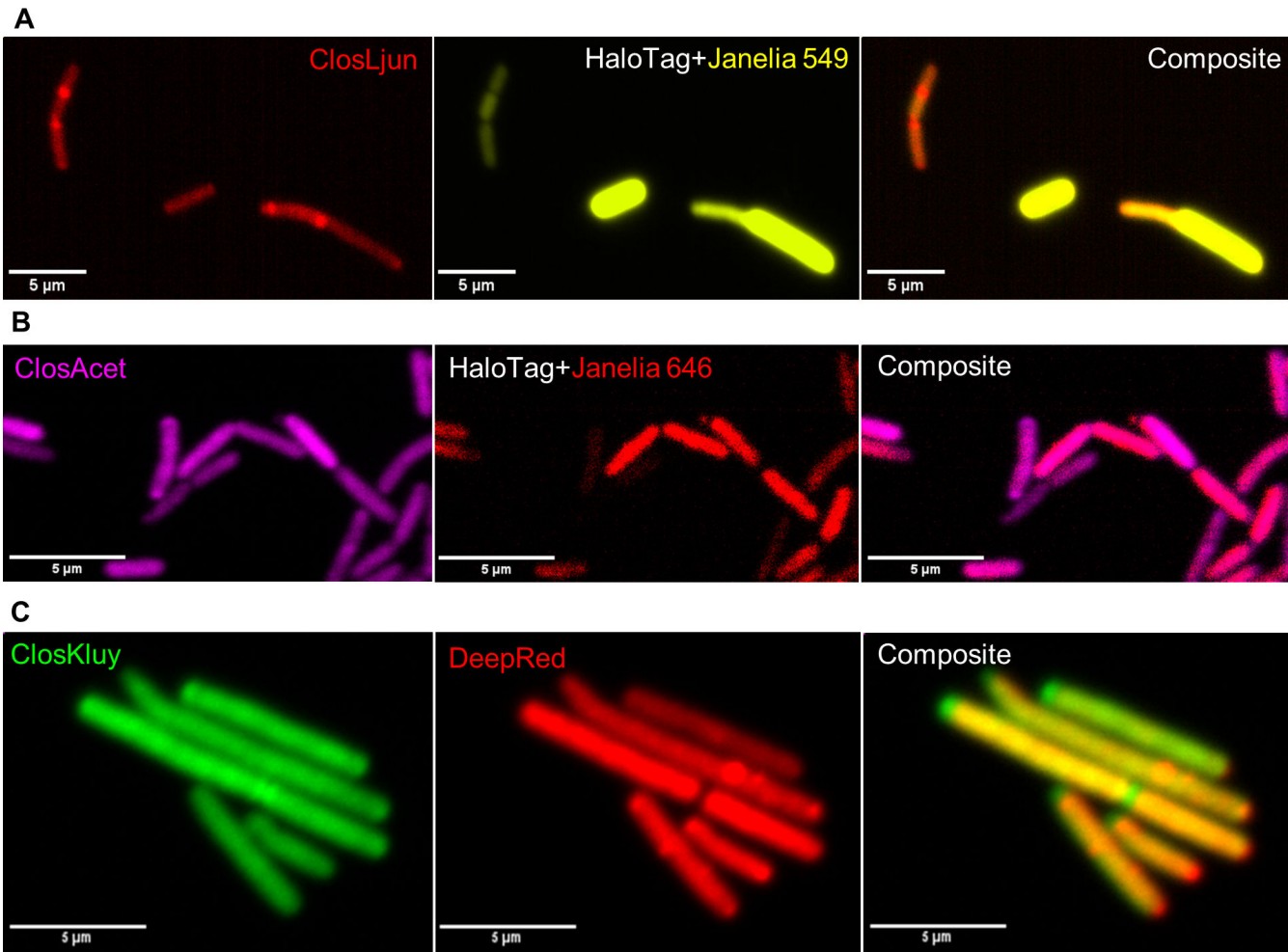

**FIG 2** rRNA is compatible with common fluorescent protein labeling techniques. (A) C. *ljungdahlii*-p100ptaHalo (*Clj*-ptaHALO, top row) is labeled with ClosLjun and Janelia 549, a yellow-emitting ligand for HaloTag. (B) *C. acetobutylicum*-p100ptaHalo (*Cac*-ptaHALO, bottom row) is labeled with ClosAcet (pseudo-colored magenta) and Janelia 646, a far-red ligand for HaloTag. (C) *C. kluyveri*'s proteins were labeled with CellTracker Deep Red since there have been no reports of successful exogenous gene expression in the organism. This dye was also found to be compatible with in-solution rRNA-FISH.

is commitment to sporulation and the second is a shrinking viable cell population (29). As previously mentioned, rRNA-FISH labeling is correlated with active cell growth and translation, so we hypothesized that the labeled fraction of the population would decrease in later stationary phase as cells commit to sporulation and the viable fraction decreased. We performed three batch cultures of *C. acetobutylicum* and tracked $OD_{600}$ and the fraction of the population which could be labeled by rRNA-FISH (Fig. 3). The labeled fraction sharply declined at the onset of stationary phase, which corresponds to the increase of cells committing to sporulation and at the same time to a decrease in viable cell fraction which is typical of *C. acetobutylicum* batch cultivation. We would conclude that rRNA-FISH can be used to assess the growth (cell expansion) potential of a culture, but not strictly speaking, cell viability.

## *C. ljungdahlii* and *C. kluyveri* exchange rRNA and form rRNA hybrid cells

Several attempts to produce $C_4$ to $C_8$ alcohols and carboxylates from $C_1$ gasses (e.g., CO and $CO_2$) have been reported in the literature (30). Most were based on the co-cultivation of an acetogen (e.g., *C. ljungdahlii* [11], *Clostridium autoethanogenum* [31], *Clostridium aceticum* [32], or *C. carboxidivorans* [16]) with *C. kluyveri*. In those systems, the acetogen consumes $CO/CO_2/H_2$ mixtures (syngas) to produce ethanol and acetate, which

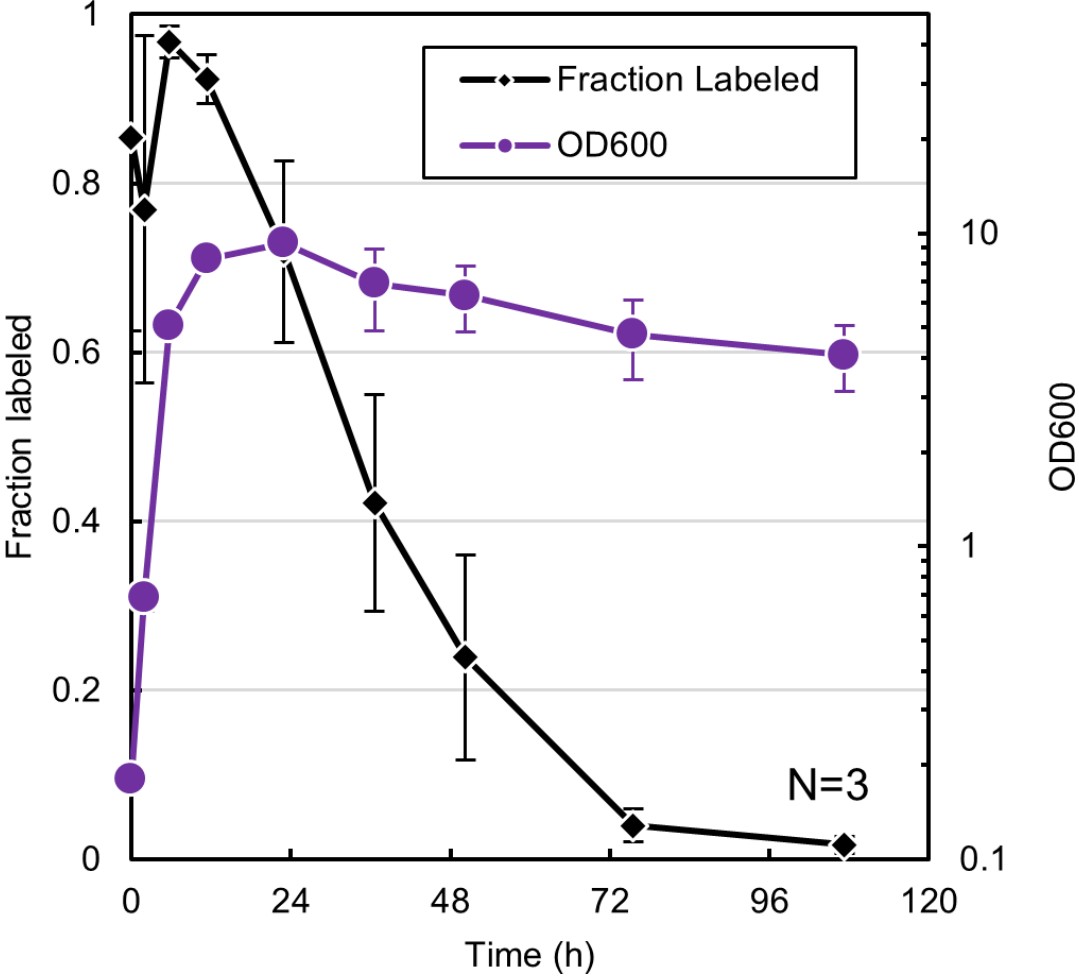

**FIG 3** rRNA-FISH as an indicator of culture health/activity. $OD_{600}$ and fraction of the population which was labeled by rRNA-FISH during batch cultivation of *C. acetobutylicum* in triplicate. Cells were labeled with ClosAcet. Cells with a fluorescent signal brighter than background were deemed "labeled." Error bars represent a single standard deviation above and below the average.

is further converted to butyric and caproic acid by *C. kluyveri*. Since the primary focus of such systems has been on metabolic production, interspecies interactions have been largely ignored. Diender et al. report that the presence of *C. kluyveri* redirects metabolic fluxes in *C. autoethanogenum*, resulting in increased ethanol production (31). The authors conclude that thermodynamic effects caused by continuous uptake of ethanol by *C. kluyveri* contribute to the effect but do not go further.

Charubin et al. had hypothesized that heterologous cytoplasmic exchange was widespread in nature, having implicated syntrophic cross feeding as the primary impetus and recognized that syntrophic cross feeding is ubiquitous in naturally occurring microbiomes (9). Here, we hypothesized the *C. kluyveri* would be capable of forming hybrid cells with *C. ljungdahlii*, provided the culture conditions would create syntrophic interdependence. To do this, we modified the typical growth medium for *C. kluyveri* (Turbo Clostridial Growth Medium [TCGM]-Ckl) which contains ethanol (343 mM) and acetate (166 mM) as the sole carbon sources. Acetate is an essential substrate of *C. kluyveri* when the only other carbon source is ethanol (33). Acetate is also the predominant product of energy metabolism in *C. ljungdahlii* when grown on fructose and gases, but *C. kluyveri* cannot use either fructose or gasses as substrates for energy metabolism. For the co-culture, we modified TCGM-Ckl by removing acetate entirely from the medium, adding 11 mM fructose and pressurizing the headspace with 35–45 psi of $H_2/CO_2$. In so doing, *C. kluyveri* becomes dependent on acetate formation from *C.*

*ljungdahlii*. We performed three biological replicates with a starting ratio of roughly 3:1 *C. ljungdahlii* to *C. kluyveri*. Samples were collected over 36 h, labeled with ClosLjun and ClosKluy, and analyzed using flow cytometry and microscopy. The growth of the culture (Fig. 4A), the relative subpopulation fraction of each species (Fig. 4B), and the prevalence of "hybrid events" (Fig. 4C) were tracked. In this case, events with both ClosLjun and ClosKluy fluorescence constitute a "hybrid event," since the cell contains rRNA from both organisms. During the first 12 h of growth, the $OD_{600}$ roughly quadrupled, but the ratio of *C. ljungdahlii* to *C. kluyveri* remained relatively constant. At that point, the frequency of unidentifiable cells ("unID," Fig. 4B) increases sharply, indicative of declining culture health. The unID cells represent cells that do not grow actively since they contain low concentrations of ribosomes/rRNA, thus displaying low translation activity. The "hybrid events" have strong green fluorescence from ClosKluy and strong red fluorescence from ClosLjun. The frequency of "hybrid events" reached its highest at 3.6 h, then decreased as culture health declined. The samples used for flow cytometry were then examined by confocal microscopy. Select images from the first three time points are presented in Fig. 4D through G. Ribosomal distribution was not always homogeneous, but rather,

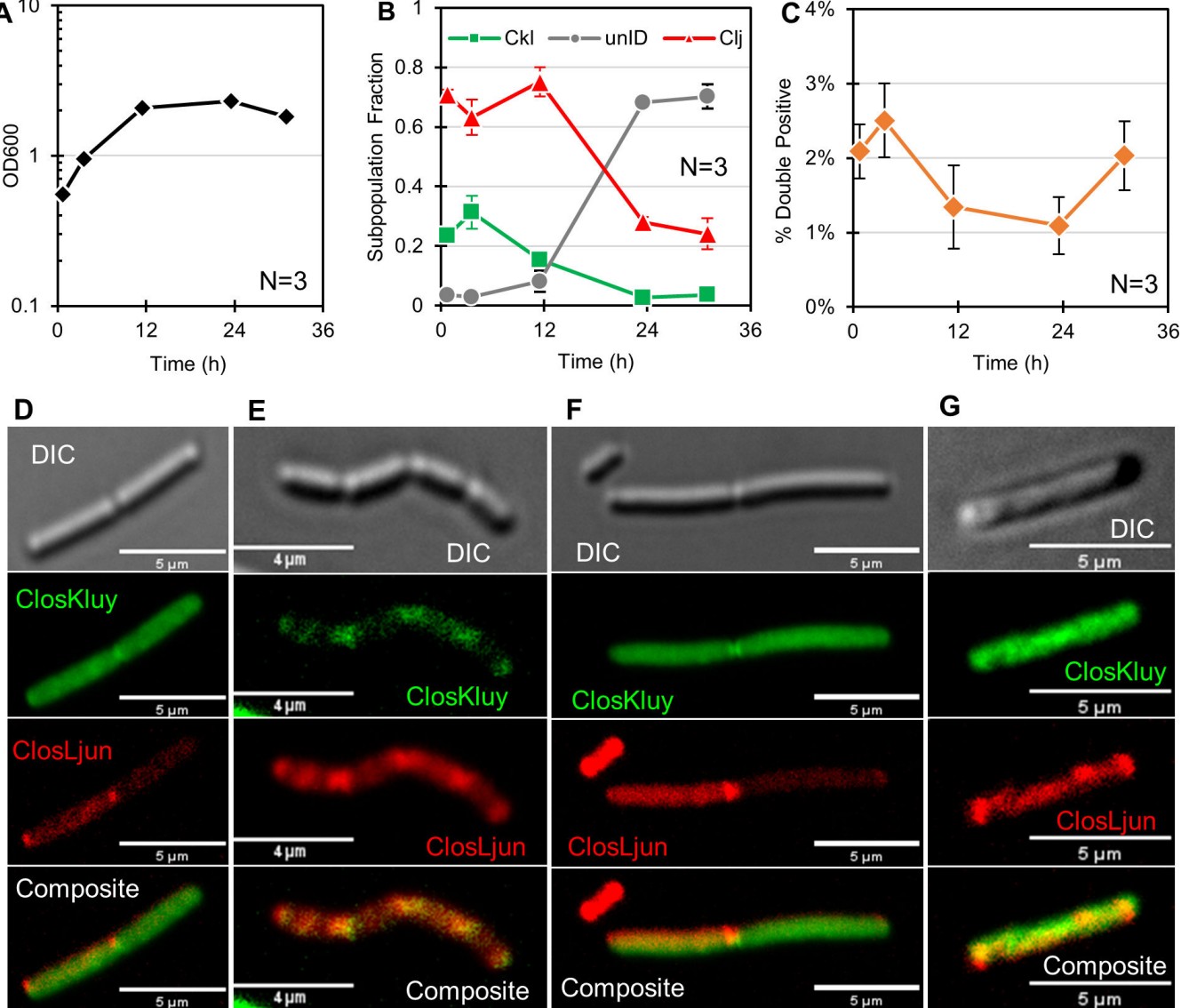

**FIG 4** rRNA hybrids in a co-culture of *C. ljungdahlii* (Clj) and *C. kluyveri* (Ckl). (A) The $OD_{600}$ of three biological replicates. (B) The subpopulation fractions. (C) The hybrid cell frequency as a percentage of total. A hybrid cell from the 1-h time point (D) from the 4-h time point (E and F) and from the 12-h time point (G).

in some cases, localized in subcellular regions. Interestingly, the two ribosomal probe's signals overlapped in some cells (Fig. 4E) but not other cells (Fig. 4F), and this is likely the result of the microscopy images only capturing a moment in the lifespan of a hybrid cell. We suspect that the exchange and distribution of ribosomes is governed dynamically by unknown mechanisms that are characteristic of the parental strain type. This prompted us to carry out the ribosomal localization studies later in the manuscript. Cells with signal from both probes were found at roughly the frequency that flow cytometry predicted: approximately one to two cells on a random frame of about 100 cells.

## rRNA-FISH tracks the population dynamics of a triple co-culture and identifies binary fusion events

To demonstrate the multiplexing ability of ClosAcet, ClosKluy, and ClosLjun, we applied our hybridization protocol to a synthetic consortium of *C. acetobutylicum, C. ljungdahlii*, and *C. kluyveri* and mono-culture controls. The optical density of the cultures is shown in Fig. 5A, and the subpopulation dynamics for the triple co-culture are shown in Fig. 5B. In this case, the preculture and culturing conditions were such that *C. acetobutylicum* were suboptimal, and this could be determined by rRNA-FISH. There is agreement between the growth behavior of the mono-cultures and those in the triple culture. Both *C. kluyveri* and *C. ljungdahlii* grew substantially in the mono-cultures, but *C. acetobutylicum* did not. This pattern is reflected in the subpopulation dynamics, insomuch as *C. kluyveri* and *C. ljungdahlii* become the predominant species and *C. acetobutylicum* becomes the minor population. The rRNA-FISH data also suggest large numbers of non-actively translating cells, indicated by the large unidentifiable population (i.e., "unID" in Fig. 5B). In healthy, exponentially growing cultures, the fraction of unlabeled cells rarely rises above 10% (discussed in next section), but the fraction of unlabeled cells in this culture was roughly 40% during growth.

We also tracked the prevalence of the binary hybrid populations via flow cytometry (Fig. 5C). *C. kluyveri-C. ljungdahlii* hybrids were more common than hybrids containing *C. acetobutylicum*. This may be in part due to the poor health of *C. acetobutylicum,* but the motivations and mechanism of heterologous cell fusion are not fully understood. Microscopy was performed to capture the hybrid events (Fig. 5D through F). A cluster of *C. kluyveri-C. ljungdahlii* hybrids is presented in Fig. 5D. In this cluster, nearly all cells contain the ClosLjun probe, but some contain the ClosKluy probe. This suggests that the green fluorescence is not merely a fluorescent artifact, but rather that it comes from ClosKluy-labeled *C. kluyveri* rRNA that co-exists with *C. ljungdahlii* rRNA. Since all cells in this cluster and neighboring cells experienced an identical hybridization environment and handling, we would expect to see identical labeling if *C. kluyveri* rRNA was not actually present. Moreover, the spatial pattern of ClosKluy and ClosLjun labeling is nearly identical, suggesting that the translation machinery from both organisms is localized in the same cellular compartments. It is possible that rRNA from both organisms form actively translating ribosomes. The peculiarity of morphology prompted the following section's investigation into the characteristic ribosomal localization patterns of each organism. Figure 5E and F presents *C. acetobutylicum-C. kluyveri* hybrids which were more difficult to find. These events are similar to those presented in previous work from our group (6). Finally, a small biofilm fragment was isolated from the labeled culture and imaged to demonstrate the potential of rRNA-FISH to deconvolute complex interspecies structures (Fig. 5G). Full-frame images are available in the supplemental material (Fig. S9 to S11)

## Characterization of cellular rRNA localization in *C. kluyveri, C. acetobutylicum*, and *C. ljungdahlii*

Unlike well-studied microorganisms such as *E. coli* and *Bacillus subtilis*, subcellular organization has not been extensively studied in *C. kluyveri*, *C. acetobutylicum*, or *C. ljungdahlii*. Many prokaryotic species have a nucleoid: a cytoplasmic region containing condensed chromosomal material and transcription machinery. The nucleoid occupies

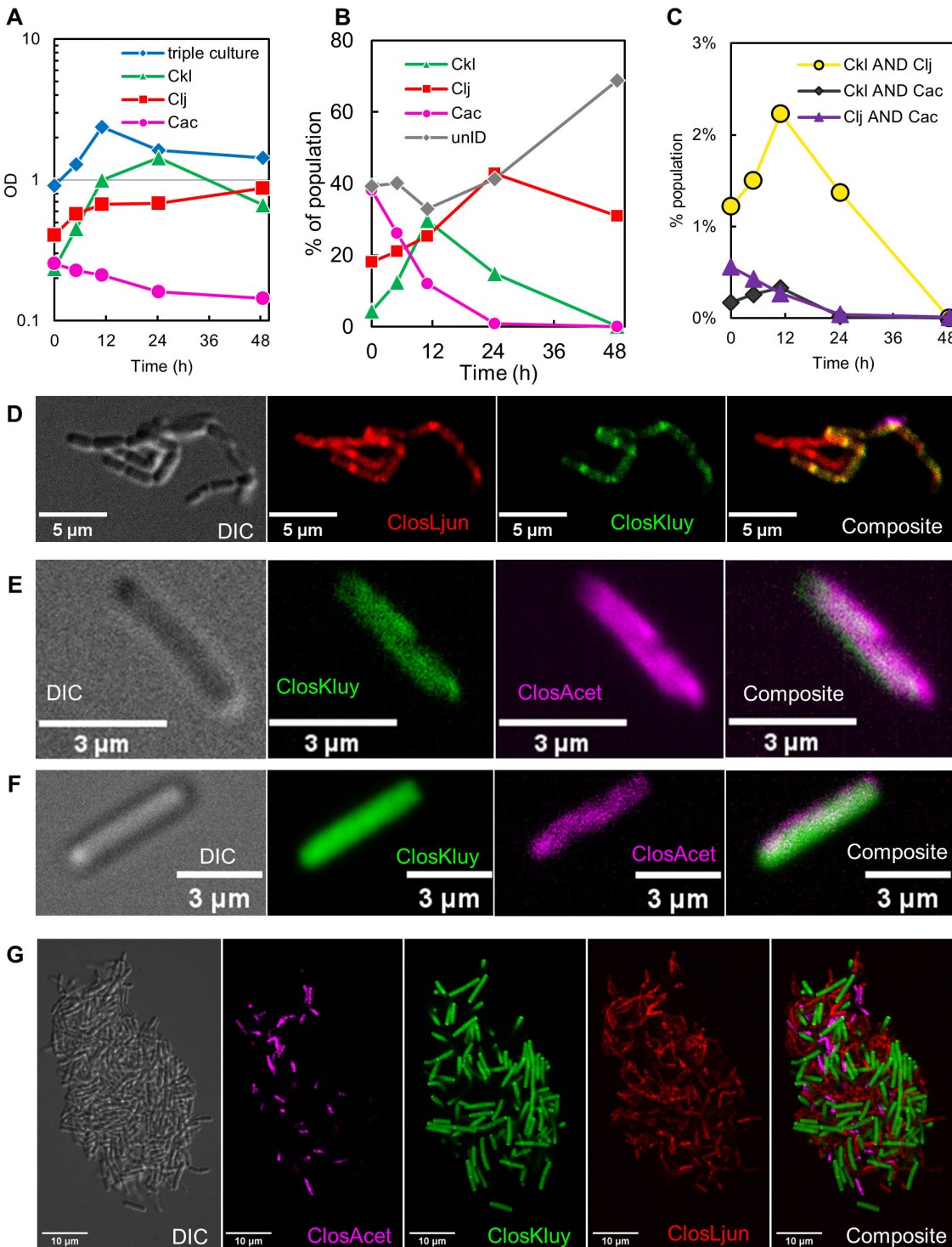

**FIG 5** rRNA-FISH applied to a triple culture of *C. acetobutylicum*, *C. kluyveri*, and *C. ljungdahlii* and mono-culture controls. (A) The optical density of the triple culture and mono-culture controls. (B) The subpopulation dynamics of the three organisms and unidentifiable cells ("unID") which are deemed inactive since they lack sufficient translation machinery to be clearly labeled by rRNA-FISH. (C) The frequency of the hybrid populations as measured by flow cytometry. (D) A cluster of cells containing *C. ljungdahlii-C. kluyveri* hybrid cells. (E and F) *C. acetobutylicum-C. kluyveri* hybrids. (G) A biofilm fragment containing all three species isolated from the culture.

the center of the cell and excludes the translation machinery. An important characteristic of each species is the ratio of the nucleoid size to the cytoplasm size which is known

as the N/C ratio. Generally, the nucleoid size does not scale proportionally with cell size. So, the nucleoid of a larger cells occupies a smaller fraction of the cytoplasm (i.e., larger cells have smaller N/C ratios) (34). In *E. coli*, the strength of nucleoid exclusion was inversely related to the N/C ratio (34). From this, the authors concluded that larger cells, such as Firmicutes (which have typically low N/C ratios), would tend to exhibit stronger exclusion of translation machinery (34). Indeed, *B. subtilis*, the model organism for Firmicutes, exhibits strong nucleoid exclusion effects (35, 36). Our rRNA-FISH probes target the 23S rRNA of the 50S large ribosomal subunit. Virtually all of the rRNA is incorporated into functional ribosomal subunits. Only about 2%–5% of rRNA is found in "ribosomal precursors," which are the immature ribonucleoprotein complexes which have yet to become a functional ribosome (37), and there are no free rRNA in the cells (38). That the transcription and translation machineries are separated apparently challenges the well-established paradigm in microbial biology that translation initiation occurs on the nascently forming mRNA transcript. It is important to distinguish between "bound" ribosomes which are a complex of a large and small subunit, mRNA, and the nascent polypeptide chain, and "unbound" ribosomes which are simply the individual subunits. About 80% of ribosomes are actively translating (39), and this is important in light of work which demonstrates that while the majority of translation occurs outside the nucleoid, unbound ribosomal subunits freely diffuse throughout the cell (40). So, in essence, rRNA-FISH fluorescence generally marks areas of active translation. While this notion has been challenged (41) for mixed environmental microbial populations due to the variable physiological characteristics of the diverse microbial populations in environmental samples, in pure cultures and defined synthetic consortia, published data (e.g., reference 36) and our data below support this assertion. One would conclude that compartmentalized rRNA-FISH fluorescence represents the compartmentalization of actively translating ribosomes.

We observed the most distinct ribosomal localization and striking division behavior in *C. ljungdahlii* (Fig. 6). *C. ljungdahlii* maintained exponential growth for about 20 h after inoculation, during which fluorescence was high at the population level (Fig. 6A and B). At $OD_{600}$ of 0.3, corresponding to t3 (Fig. 6A), most cells formed elongating "chains" of replicating but not separating cells. Differential interference contrast (DIC) images clearly show the nascent cleavage furrow forming throughout the chain at regular intervals (Fig. 6C; Fig. S12). For each microscopic image, the full-frame image is provided in the supplemental figures to support the argument that the cells provided in the main figures are representative/typical for the entire population. The plots to the right of the microscopy images present the fluorescent intensity along the major axis of the bacilliform-cell chain going from left to right. The fluorescent intensity is plotted in arbitrary units so as to compare the relative fluorescence (i.e., the prominence) of the signal within a cell, but not between cells. Microscopy of stationary phase *C. ljungdahlii* (Fig. S13) confirms the expected, canonical drop in fluorescence (Fig. 6B), which here, given the lack of *C. ljungdahlii* sporulation under these culture conditions, is likely caused by carbon starvation and a rapid decrease in the number of viable cells. One of the few fluorescent cells at the t5 stationary-phase point is shown in Fig. 6D, but the prominence of the fluorescent peak at the cleavage furrow is decreased.

Ribosomal compartmentalization can be seen in *C. kluyveri*, though the effect is less pronounced, and the phenomenon does not appear universally even during exponential phase. Four exponentially growing cells and their fluorescent profile are shown in Fig. 7C and Fig. S14. Cell 3 has distinct fluorescent puncta at the poles, labeled α and β. The translation machinery of the other cells (cells 1, 2, and 4) is more homogeneously distributed. Unlike *C. ljungdahlii*, *C. kluyveri* (which does not sporulate either under these culture conditions) remains fluorescent into late stationary phase (Fig. 7D; Fig. S15). This observation suggests that *C. kluyveri*, with its unique substrate requirements, has the ability to maintain an active translation machinery even late in stationary phase, persisting in natural milieus to scavenge its low-energy content substrates ethanol and acetate.

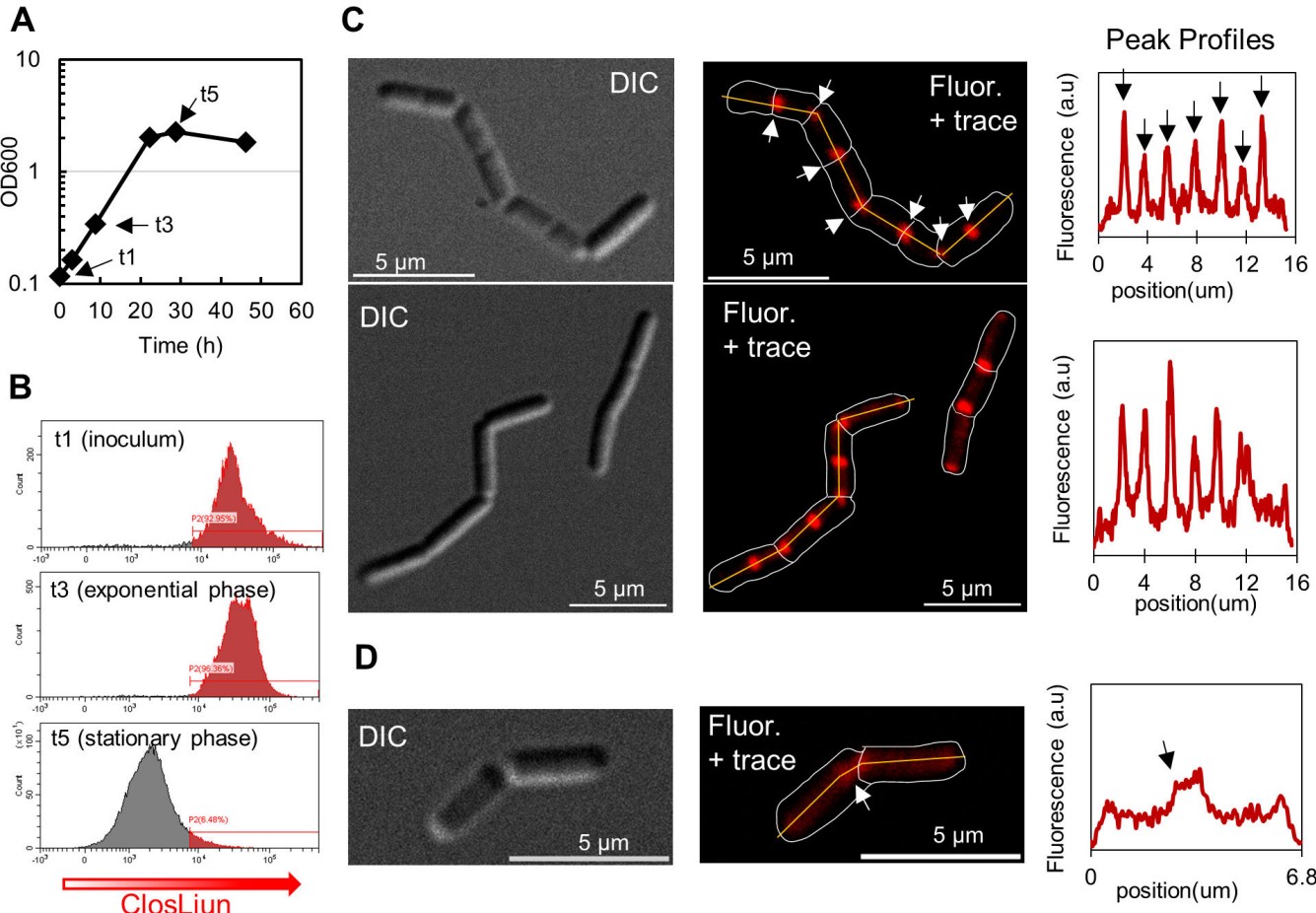

**FIG 6** rRNA localization and "chaining" in exponentially growing *C. ljungdahlii*. (A) The OD$_{600}$ and (B) flow cytometric data demonstrating a steep drop-off in fluorescence once the cells enter stationary phase. (C) DIC images of a typical *C. ljungdahlii* at early stationary phase (t3), which clearly show distinct cells bodies which have not undergone cleavage. Trace and fluorescent images show localization at the cleavage furrow and at the mid-cell. The fluorescent profile plot of the agglomerate's major axis is indicated by the yellow line in panel C. Fluorescent peaks are extremely prominent during exponential phase. (D) Very few cells imaged microscopically from stationary phase (t5) displayed fluorescence (Fig. S13). The brightest cell from these images cell was selected and analyzed as cells in panel C but had decreased peak prominence compared to exponential phase.

The surprising behavior is that of *C. acetobutylicum* (Fig. 8). Exponential-phase cells (OD$_{600}$ = ~4) were passaged into fresh media and went quickly in exponential growth as evidenced by the straight growth line on the semi-log OD$_{600}$ graph (Fig. 8A). As a population, the cells in the inoculum were very bright (Fig. 8B). Individually, cells exhibited a uniform distribution of translation machinery as evidenced by confocal microscopy and the accompanying fluorescence intensity plots (Fig. 8C; Fig. S16). To varying extents, weakly fluorescent puncta and "banding" can be found in some cells, but these represent a minority of cells and are not representative. Two hours after passaging, the fluorescent intensity of the cells as measured by flow cytometry had decreased sharply, giving rise to a second non-fluorescent population of presumably translation-inactive cells (Fig. 8B). Full-frame images in Fig. S17 and S18 show that roughly half of cells have no fluorescent signal. This was not expected as one assumes that during exponential cell growth, cells are viable and actively translating proteins, though there appears to be a larger discrepancy between CFU/mL and cell counts at early exponential phase than at mid-exponential phase (28). This is a novel observation and perhaps unique among the three *Clostridium* species examined here. Among fluorescent cells, the ribosomes had localized to the poles and the mid-cell region (Fig. 8D; Fig. S17 and S18). During early stationary phase, the cells became appreciably

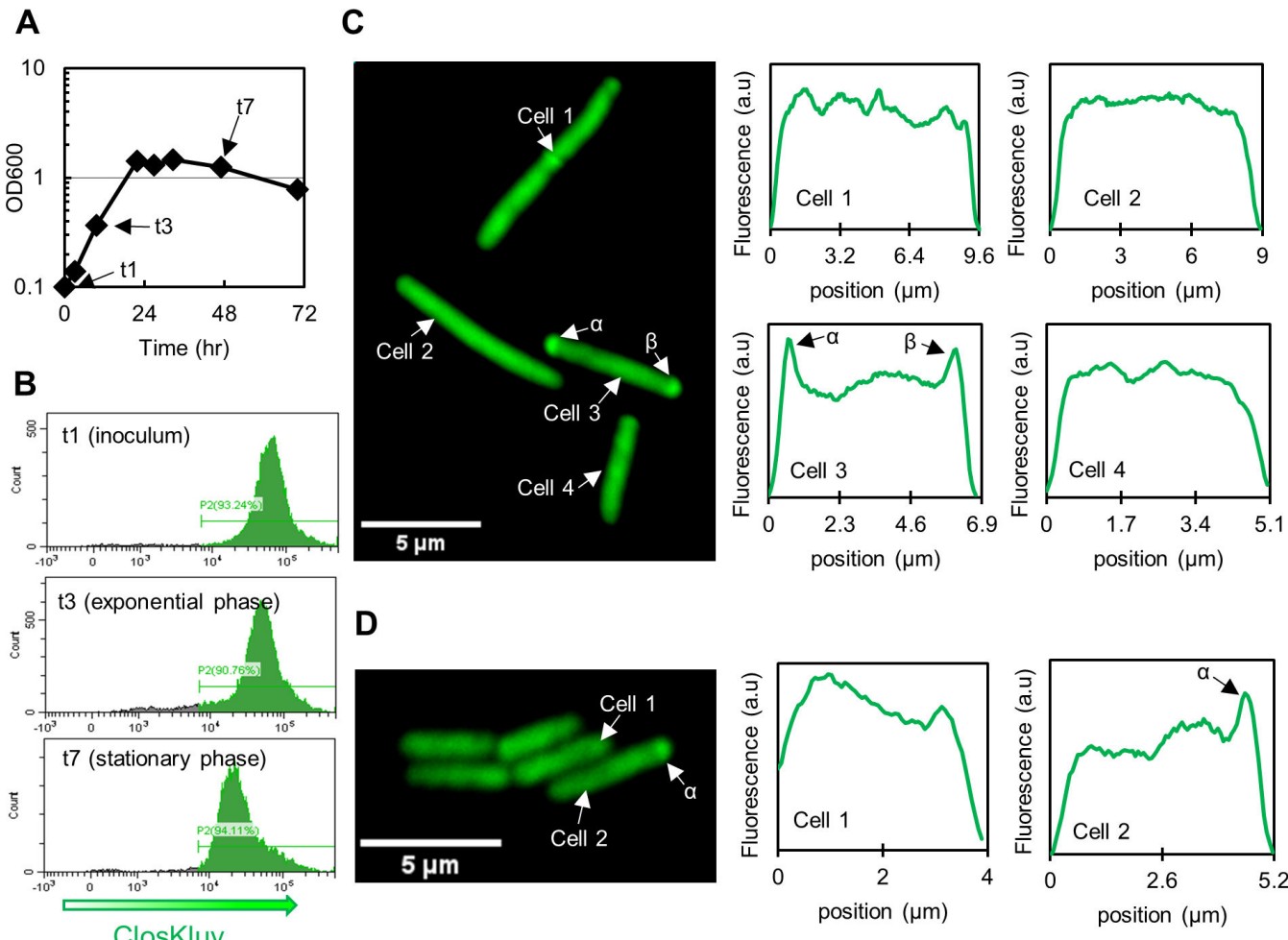

**FIG 7** rRNA localization in *C. kluyveri*. (A) The $OD_{600}$ and (B) flow cytometric data demonstrating a modest decrease in fluorescence once the cells leave exponential phase. (C) Cells exhibited ribosomal localization during exponential growth (t3) to varying extents epitomized by the four cells in this grouping. Cells 1, 2, and 4 show very little localization patterns, but cell 3 has distinct puncta at its poles, suggesting ribosomal localization is a transient phenomenon in *C. kluyveri*. (D) In late stationary phase (t7), the cells remained highly fluorescent, unlike in *C. ljungdahlii,* but mostly lacked fluorescent peaks.

brighter with more uniformly distributed fluorescence, resembling cells in the inoculum, though the most prominent peaks were still located at the poles and some banding can be seen (Fig. 8E; Fig. S19).

## DISCUSSION

Even though rRNA-FISH has been applied to prokaryotic biology for over 30 years, the development of species-specific rRNA-FISH probes in *Clostridium* spp. is somewhat rare. We attempted to use probeBase to determine the extent to which species-specific rRNA-FISH probes had been designed in among Clostridia (42), but the list seems incomplete since it does not include some of the probes listed from other sources (12). We were able to find probes designed to target *Clostridium leptum* (43, 44), *Clostridioides difficile* (45, 46), and *Clostridium perfringens* (45). We could only find two examples of species-specific probes being applied to industrially interesting Clostridia. A set of probes for *C. kluyveri*, "KCLZ," was designed to identify the species in samples from "pit mud" in the Luzhou Liquor Manufacturing Facility (47). As previously mentioned, ClosKluy and ClosCarb had been recently designed to analyze *C. kluyveri* and *C. carboxidivorans*, respectively (21).

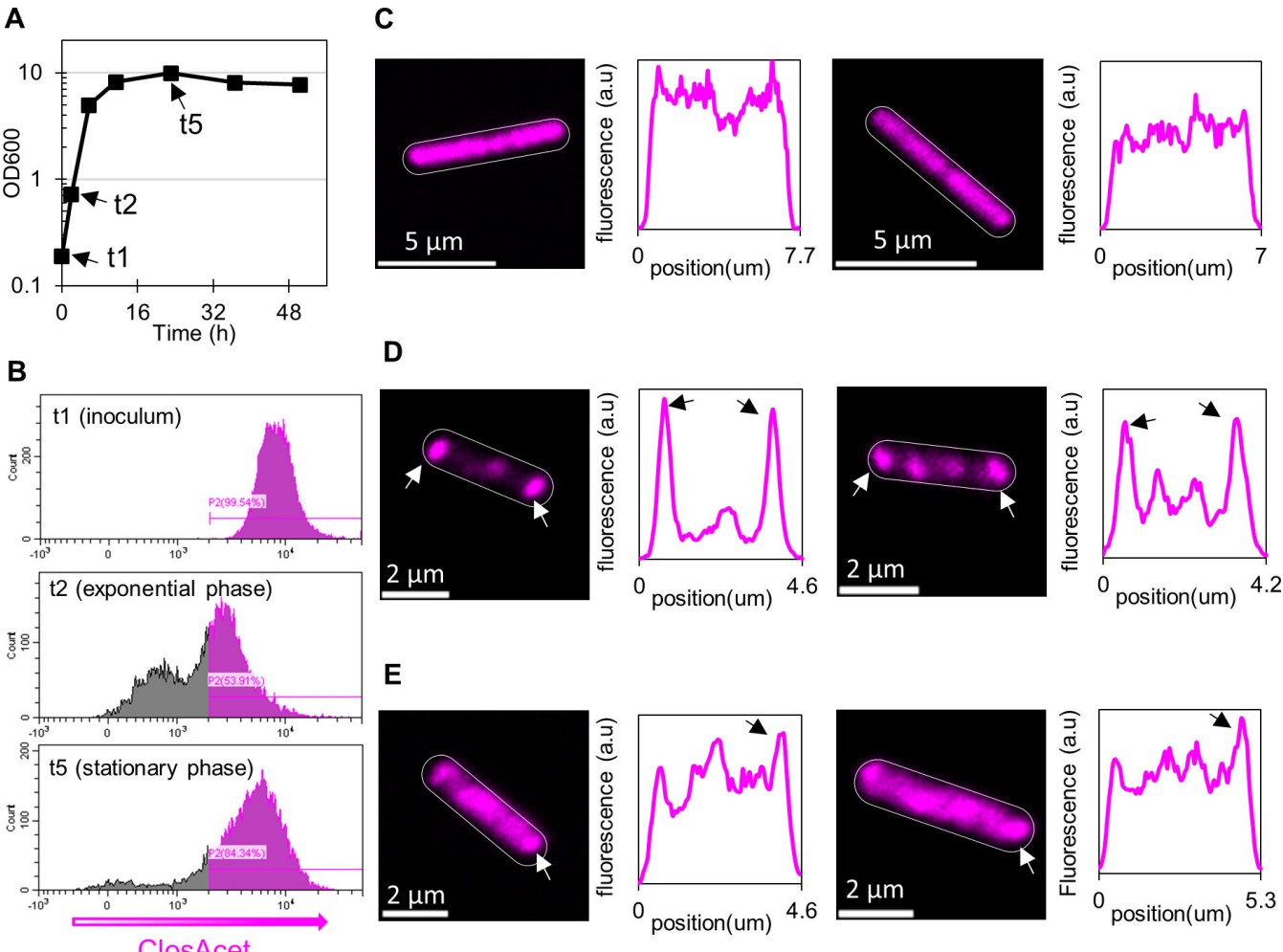

**FIG 8** rRNA localization in *C. acetobutylicum*. An inoculum grown from a colony to an OD$_{600}$ of ~4. Our rRNA-FISH method was used to analyze the inoculum (t1), exponential (t2), and stationary phase (t5) of the culture. (A) The OD$_{600}$. (B) Flow cytometry histograms of fluorescence in the population show a decrease in overall fluorescence during early exponential phase, but sustained fluorescence in early stationary phase. (C) Microscopy images and fluorescent profile plots of typical inoculum cells show no localization. (D) Microscopy images and fluorescent profiles plots of typical exponential-phase cells indicate strong ribosomal localization. (E) Microscopy images and fluorescent profile plots of *C. acetobutylicum* in early stationary phase.

Owing to its accessibility and applicability, rRNA-FISH is suitable for rapid adoption in the field of synthetic consortia microbiology. From our experience, the PROBE_DESIGN tool in ARB is straightforward and free to use. The probes are easily synthesized through commercial vendors at low cost. Additionally, a wide array of fluorophores is available allowing greater flexibility in experimental design. In our case, it was important to select probes which could be hybridized under similar conditions to allow for multiplexing, so we selected probes with similar melting temperatures as determined by freely available tools such as "OligoAnalyzer." In addition to cytoplasmic exchange studies, rRNA-FISH has several advantages over other methods. It can be used for subpopulation tracking (Fig. 4 and 5) (16, 48) as well as for identifying the fraction of actively translating (and thus actively metabolizing) cells, which, as we have shown here (Fig. 6 to 8), is very different for different species. Fluorescent reporter proteins could be used to identify cells but require genetic modifications, which are not currently available or practical in many interesting strains such as *C. kluyveri*. Dyes could be used to label the populations initially but are diluted as the cells grow. qPCR has been used to quantify subpopulations in other co-cultures (5), but provides no information about the health or expansion potential of the populations and is tedious when temporal population profiles are

desirable. Modeling frameworks aiming to predict subpopulation dynamics mathematically have been proposed (49–52). So far, these approaches are only applicable in a qualitative sense, so the collection of reliable and quantitative experimental data is paramount. In future work, the inclusion of 4′,6-diamidino-2-phenylindole (DAPI), or other DNA stains, in addition to rRNA-FISH, could be used to identify the presence of dividing cells, providing further information about the relative rates of growth and death in stationary-phase cultures (53). The ability to quickly obtain information regarding the population ratios, viability, and subpopulation activity would prove useful as a diagnostic tool for both synthetic consortia (54) and mixed fermentations (55, 56) in the industrial biotechnology setting

We explored cytoplasmic material exchange via rRNA-FISH probes between *C. acetobutylicum*, *C. ljungdahlii,* and *C. kluyveri,* identifying for the first time cellular exchange (rRNA evidence-based hybrids) between *C. ljungdahlii* and *C. kluyveri* (Fig. 4) and also strengthening the evidence for *C. acetobutylicum-C. kluyveri* hybrids. The rRNA hybrids that arose in the *C. kluyveri-C. ljungdahlii* co-cultures had a large degree of physiological heterogeneity, sometimes resembling *C. ljungdahlii*, sometimes resembling *C. kluyveri*, and sometimes resembling neither. It was important to perform the ribosomal localization studies of each species in mono-culture to better interpret the physiology of hybrid cells. Information regarding the typical, parental physiology of cells involved in cytoplasm exchange can be combined with fluorescent reporters (7, 9). The physiology of cells in Fig. 4E is characteristic of *C. ljungdahlii*. Under our experimental conditions, "chaining" was only observed in *C. ljungdahlii* (Fig. 6), and that is apparently retained in some of the *C. ljungdahlii-C. kluyveri* hybrid cells (Fig. 4E). Additionally, the localization pattern of ClosKluy- and ClosLjun-labeled rRNA resemble *C. ljungdahlii*. Distinct puncta are associated with the nascent cleavage furrow between cells (Fig. 4E and 6C).

In the context of the ever-increasing interest in microbiome studies, there is so far little attention paid to understanding the fundamentals of complex biological interactions such as those discussed here involving the exchange of cellular material. Furthermore, there is now increasing interest regarding the role and regulation of physical intercellular interactions in co-cultures of biotechnological interest (57). Moreover, if this exchange of cytoplasmic material, especially genetic material, is widespread in nature, then it would greatly affect our current understanding of evolutionary ecology in clostridial microbiomes and more broadly. While this work focuses on co-cultures involving non-pathogenic acetogens and other *Clostridium* spp., one would expect that this behavior will likely be encountered in pathogenic *Clostridium* spp. as well. Notable is the case of the notorious pathogen *Clostridioides difficile,* also an acetogen, which will likely behave, in some respects, similarly to *C. ljungdahlii*. *C. difficile* is the leading cause of infectious diarrhea in hospitals and the community. Its infections present a significant mortality threat and thus constitute a tremendous burden on healthcare systems worldwide (58). Understanding interspecies relationships among pathogenic and non-pathogenic *Clostridium* spp. and other organisms would provide far-reaching insights into agriculture and human health, as well as industrial fermentations (59).

## MATERIALS AND METHODS

### Probe design

All probes used in this study can be found in Table S3. The ClosAcet and ClosLjun probe sets were designed and named similarly to Schneider et al., who demonstrated an analogous system on *C. carboxidivorans* and *C. kluyveri* (21). Briefly, the PROBE_DESIGN tool in ARB (Latin, _arbor_ meaning tree) software (version 7.0) (60) was used with the SILVA (Latin, _silva_ meaning forest) database LSU Ref NR 99 (version 138.1) library (61) to generate potential probe sequences (https://www.arb-silva.de/). ClosAcet was designed to target all entries within the *C. acetobutylicum* ATCC 824, DSM 1731, and EA2018 subtree. Designing a probe which labeled only *C. ljungdahlii* (and not the closely

related *C. autoethanogenum*) was deemed impossible, since the only suitable probe coincidentally had specificity for *C. acetobutylicum*. Therefore, ClosLjun was designed with specificity to *C. autoethanogenum* DSM 10061 and *C. ljungdahlii* DSM 13528. Probe selections were made based on the ARB's estimated selectivity of the probe (i.e., "equal"). We intended to label multiple species simultaneously, so it was essential that the ideal hybridization stringency (determined by formamide concentration in the hybridization buffer) was similar between all probe species. Under optimal hybridization conditions, enough formamide was added to prevent imperfect duplex formation (off-target labeling), but overly stringent hybridization conditions prevented desirable bonding between probe and rRNA target (14). We hypothesized that probes with similar melting temperatures would share optimal hybridization conditions. ARB provides a basic melting temperature based on probe length and GC content, but nearest-neighbor (NN) methods are far more accurate and parameters specifically for RNA/DNA duplex formation are readily available (62). We used the "OligoAnalyzer" tool from Integrated DNA Technologies (IDT) to select probes with melting temperatures between 68°C and 72°C, presuming a 1 mM probe concentration and 900 mM $Na^+$ concentration. (https://www.idtdna.com/calc/analyzer). Cy3 was chosen as the fluorescent tag for ClosAcet, and Cy5.5 was chosen as the fluorescent tag for ClosLjun. These fluorescent molecules are known to be orthogonal green fluorophores (such as Alexa Fluor 488, used with ClosKluy and the OregonGreen HaloTag Ligand) on the fluorescent channels used for flow cytometry and microscopy. Orthogonality was checked with the fluorescent spectrum tool from AAT BioQuest (https://www.aatbio.com/fluorescence-excitation-emission-spectrum-graph-viewer). All probes were synthesized by Integrated DNA Technologies.

## Microorganisms and growth media

Mono-cultures of *C. acetobutylicum* (ATCC 824), *C. ljungdahlii* (ATCC 55383), and *C. kluyveri* (ATCC 8527) were grown in TCGM supplemented with species-specific carbon sources and nutrients as previously described (5, 6). The TCGM base stock contains the following amounts of each component per liter: yeast extract, 5 g; asparagine, 2 g; 50× mineral stock solution, 20 mL; 100× trace elements solution, 10 mL; phosphate buffer solution, 10 mL; 100× Wolfe's vitamins, 10 mL; 500× para-aminobenzoic acid (PABA) solution, 2 mL. The 50× mineral stock solution contains the following amount of each component per liter: NaCl, 50.5 g; $MgSO_4$, 17.4 g; $(NH_4)_2SO_4$, 100 g; sodium acetate, 123 g. The 100× trace elements solution contains the following amount of each component per liter: nitrilotriacetic acid, 1.5 g; $MnSO_4 \cdot 7H_2O$, 1.5 g; $FeSO_4 \cdot 7H_2O$, 1.1 g; $CoCl_2 \cdot 6H_2O$, .1 g; $CaCl_2$, 2.1 g; $ZnSO_4 \cdot 7H_2O$, o.18 g; $CuSO_4 \cdot 5H_2O$, 10 mg; $KAl(SO_4)_2 \cdot 12H_2O$, 20 mg; $H_3BO_3$, 10 mg; $Na_2MoO_4 \cdot 2H_2O$, 10 mg; $NiCl_2 \cdot 6H_2O$, 30 mg; $Na_2WO_4 \cdot 2H_2O$, 20 mg; $Na_2SeO_3 \cdot 5H_2O$, 300 µg. In addition, 100× Wolfe's vitamins contains the following amounts per liter: pyridoxine-HCl, 10 mg; thiamine-HCl, 5 mg; riboflavin, 5 mg; calcium pantothenate, 5 mg; thioctic acid, 5 mg; PABA, 5 mg; nicotinic acid, 5 mg; vitamin $B_{12}$, 100 µg; D-biotin, 2 mg; folic acid, 2 mg. The phosphate buffer contains the following amounts per liter: $KH_2PO_4$, 100 g; $K_2HPO_4$, 125 g. The phosphate buffer is adjusted to a pH of 6.9 with NaOH. The 500× PABA solution contains 2 g/L PABA. The 100× phosphate buffer, 100× Wolfe's vitamins, and 500× PABA are filter sterilized (0.2 µm). Yeast extract, asparagine, 100× trace elements, and 50× mineral solution are dissolved in 800 mL RO-$H_2O$ and autoclaved at 121°C for 30 min. After cooling, 100× phosphate buffer solution, 100× Wolfe's vitamins, and 500× PABA solution are added. For cultures containing *C. acetobutylicum*, TCGM was supplemented with 160 mL/L of a filter-sterilized (0.2 µm) 500 g/L glucose solution and 10 mL/L of a filter-sterilized (0.2 µm) 500 g/L fructose solution (80/5 TCGM). For cultures containing *C. ljungdahlii*, TCGM was supplemented with 10 mL/L of filter-sterilized (0.2 µm) 500 g/L fructose (0/5 TCGM). For cultures containing *C. kluyveri*, TCGM was supplemented with 20 mL/L ethanol, 40 mL/L of 200 g/L sodium acetate solution, 25 mL/L of a 100 g/L sodium bicarbonate, 10 mL/L of a 15 g/L L-cysteine-HCl solution, and an additional 10 mL/L of the 100× phosphate buffer (TCGM-Ckl). The balance was sterile RO-water. All media were left to deoxygenate in an anaerobic

chamber (Forma Anaerobic System; Thermo Fisher Scientific) for at least 2 days. For growth of solid media, 2× YTG plates were used which contain the following per liter: tryptone, 16 g; yeast extract, 10 g; NaCl, 4 g; glucose, 5 g; agar, 15 g. After combining all the ingredients, the medium is adjusted to a pH of 5.8 with 5 M HCl before autoclaving. The medium is cooled to 65°C before addition of antibiotics and poured into plates immediately afterward.

## Growth of mono-cultures and precultures

Exponentially growing cultures of *C. acetobutylicum, C. ljungdahlii*, and *C. kluyveri* were diluted 9:1 in pure dimethyl sulfoxide (DMSO) and stored at −80°C (New Brunswick Scientific, Edison, NJ, USA). For *C. acetobutylicum,* 150 µL of frozen stock was streaked on 2× YTG plates anaerobically and allowed to incubate at 37°C for more than 4 days to allow for sporulation. After this, plates were stored aerobically at 4°C for up to a year to select against vegetative cells which would be killed in exposure to oxygen. To begin *C. acetobutylicum* liquid cultures, 10 mL of 80/5 TCGM is inoculated with a single colony and heat shocked for 10 to 20 min at 80°C to initiate spore germination. Growth was typically observed after 14 to 20 h. To prevent acid death, the culture's pH is adjusted to 5.2 with NaOH after 22 to 28 h, unless passaged beforehand. To begin the *C. ljungdahlii* liquid cultures, 1 mL frozen stocks of *C. ljungdahlii* was sowed anaerobically into 9 mL of 0/5 TCGM and passaged as needed after 16 h once the cell reached log phase. For *C. kluyveri*, 1 mL of cells were sowed into 19 mL of TCGM-Ckl. The plasmid of *C. acetobutylicum*-p100ptaHALO and *C. ljungdahlii*-p100ptaHALO was maintained by addition of 100 µg erythromycin (Em) per milliliter of culture media from a 1,000× concentrated stock. When growing *C. acetobutylicum* from colonies, Em was added after heat shock.

## *C. kluyveri* and *C. ljungdahlii* co-culture preparation

A special culture medium was prepared for the *C. kluyveri* and *C. ljungdahlii* co-culture by omitting acetate entirely from TCGM-Ckl and adding 4 mL/L of the 500 g/L fructose solution (TCGM-Ckl/Clj). Precultures of *C. ljungdahlii* and *C. kluyveri* (20 mL) were grown to exponential phase (OD$_{600}$ of 0.4 to 0.6) in 0/5 TCGM and TCGM-Ckl, respectively. To initiate the co-culture, 10 mL of each preculture was separately washed once in TCGM-Ckl/Clj and resuspended into 1 mL. The two 1 mL cell concentrations are then added to 13 mL of fresh TCGM-Ckl/Clj in a sealed Balch tube (18 mm × 150 mm, ChemGlass), resulting in a 15 mL co-culture with an *R* value of ~1. The headspace of the Balch tube was filled to between 35 and 45 psi of an 80/20 blend of H$_2$ and CO$_2$ gas. After each sampling, the headspace was repressurized.

## *C. acetobutylicum* and *C. ljungdahlii* co-culture preparation

The results of this co-culture are described in supplemental materials. A 90 mL culture of *C. ljungdahlii*-p100ptaHalo was grown to exponential phase (OD$_{600}$ of 0.4 to 0.6) and a 30 mL culture of *C. acetobutylicum* was grown to early stationary phase (OD$_{600}$ of 6 to 8). The full 90 mL *C. ljungdahlii*-p100ptaHalo culture was washed three times to remove residual erythromycin and concentrated to 1 mL. Five milliliters of *C. acetobutylicum* was washed and resuspended in 29 mL of 80/5 TCGM. The high-density *C. ljungdahlii*-p100ptaHalo inoculum was added to the *C. acetobutylicum* to start the co-culture (5).

## *C. acetobutylicum, C. ljungdahlii, C. kluyveri* triple co-culture preparation

To prepare the triple co-culture medium, TCGM-Ckl was supplemented with 3.5 g/L glucose and 3 g/L fructose. *C. ljungdahlii* and *C. kluyveri* were grown separately to early exponential phase (OD$_{600}$ of 0.2 to 0.4) in 40 mL cultures, and *C. acetobutylicum* was grown to early stationary phase (OD$_{600}$ of 6 to 8) in a 10 mL culture. A volume of cells corresponding to 4 OD$_{eq}$ (where OD$_{eq}$ is the product of the volume [mL] and the

$OD_{600}$) of each species is spun down and washed in the triple culture media twice and resuspended in 3.33 mL. Then the samples are combined to form a 10 mL triple culture.

## In-solution RNA-FISH protocol

rRNA-FISH was performed similarly to Baumler et. al (16). Culture samples were washed twice in filter-sterilized (0.2 µm) ice-cold 1× PBS (Gibco, pH 7.4) via centrifugation (5 min–10 min, 3,220 rcf, 4°C, Centrifuge 5810 R; Eppendorf). Then the pellet was resuspended thoroughly in ice-cold 1× PBS and fixed by 1:1 dilution in ice-cold absolute ethanol. All samples are stored at −20°C for up to a month.

For each in-solution RNA-FISH sample, 0.15 $OD_{eq}$ of cells were pelleted via centrifugation (10 min, 10,000 rcf, 4°C, Z216 MK; Hermle) in a 1.6 mL microcentrifuge tube. The $OD_{eq}$ is calculated by the following equation:

$$OD_{eq} = OD_{600} * \text{volume (mL)}$$

After carefully removing the supernatant, the pellet was dried for 5 to 15 min at 46°C (Isotemp Hybridization Incubator; Fisher Scientific) to remove residual ethanol. The pellet was resuspended in 75 µL of hybridization buffer containing 0.9 M NaCl, 0.02 M Tris-HCl (pH 7.0), 20% (vol/vol) formamide, 0.01% sodium dodecyl sulfate, and 1 µM probe(s) in aqueous solution and incubated at 46°C for 5 h, unless otherwise noted. During hybridization, 50 mL of wash buffer was prepared containing 0.215 M NaCl, 5 µM EDTA, 0.02 M Tris-HCl (pH 7.0), and 0.01% sodium dodecyl sulfate and pre-warmed to 48°C. Immediately following hybridization, the cells are pelleted via centrifugation (8 min–10 min, 10,000 rcf, Centrifuge 5418; Eppendorf), and the supernatant is discarded into as formamide waste stream. The cells are washed twice via incubation in 500 µL of washing buffer for 20 min at 48°C. Finally, the cells are washed once and resuspended in 1 mL of ice-cold 1× PBS. For the composition of hybridization and washing buffers used throughout the manuscript, refer to Table S4.

## Deep Red labeling

CellTracker Deep Red was applied as previously described (9). Briefly, 15 µg of CellTracker Deep Red was dissolved in 20 µL of DMSO to generate a 1,000× (1 mM) labeling stock. The 1,000× stock was added directly to the culture media and incubated at 37°C for 40 min. Excess dye was removed by washing the cells three times in medium via centrifugation and resuspension.

## HaloTag labeling

The HaloTag protein was labeled as previously described (26). Prior to fixing, 0.5 µL aliquots of concentrated ligand were added to 500 µL aliquots of cells and incubated at 37°C for 15 to 60 min anaerobically. The final ligand concentration for Janelia 549 and 646 was 200 nM. The final ligand concentration for OregonGreen was 1 µM. After labeling, the cells were washed twice in warm 1× PBS. Cells labeled with OregonGreen were further resuspended in fresh, warm culture media and incubated for 30 min at 37°C to wash out any residual ligand. After one final wash in 1× PBS, the cells were ready for flow cytometry, microscopy, or pre-hybridization fixing.

## Flow cytometry

A flow cytometer (CytoFLEX S, Beckman Coulter Life Sciences, Indianapolis, IN, USA) was used to collect fluorescence and light scattering data from individual cells. CytExpert (Beckman Coulter, version 2.4.0.28) was used as the acquisition and data processing software. Cells were diluted to approximately $10^5$ cells per milliliter, (ca. $OD_{600}$ of 0.1) in 1× PBS on ice. A 488 nm laser was used to collect forward-angle light scatter (FSC) and right-angle side scatter (SSC) for the sample. Only events surpassing a pulse height

mSystems

of 900 on the SSC channel were counted. The sample feed rate was adjusted between 10 and 60 mL/min to reach a sampling rate of about 1,000 events/s. In all samples, the abort rate was kept below 7% by adjusting sampling rate. For each sample, 5 µL was interrogated. The fluorescent signal of ClosKluy and OregonGreen was excited by a blue laser and filtered through a 525/40 band pass filter (BPF) before acquisition. The fluorescent signal of ClosAcet and Janelia 549 was excited by yellow laser and filtered through a 585/42 BPF before acquisition. The fluorescent signal of ClosLjun, Janelia 646, and Deep Red was excited by a red laser and filtered through a 712/25 BPF before acquisition. The gain of the avalanche photodiodes was adjusted to 125, 1,000, and 850 to acquire ClosKluy, ClosAcet, and ClosLjun, respectively. The standard gain following QC was used for the HaloTag ligands and Deep Red. For each channel, the fluorescent intensity is given by the area under the curve of emission intensity, rather than height. For each sample, the median of the population was used to determine central tendency.

## Microscopy and image analysis

Following in-solution rRNA-FISH, between 5 and 50 µL of cell sample was dropped onto either poly-l-lysine coated chambered slides (Nunc Lab-Tek 8-Well chambered slides, Thermo Scientific) or 10-well diagnostic slides (MP Biosciences, Irvine, CA, USA) and dried completely at 46°C to ensure adhesion. Cells in the chambered slides were submerged in 200 µL of cold 1× PBS prior to imaging. Diagnostic slides were rinsed in dunk tanks of cold RO-$H_2O$, then dried quickly under compressed air to remove residual water before being interred in polyvinyl alcohol mounting medium with DABCO (PVA-DABCO) (Sigma-Aldrich, Darmstadt, Germany). All microscopy was performed on an ANDOR Dragonfly spinning disk confocal microscope under a 63× or 100× oil objective. The fluorescent signal of ClosKluy and OregonGreen was excited by a blue laser ($\lambda_{ex}$ = 488 nm) and filtered through a 521/38 BPF before acquisition. The fluorescent signal of ClosAcet and Janelia 549 was excited by yellow laser ($\lambda_{ex}$ = 561 nm) and filtered through a 594/43 BPF before acquisition. The fluorescent signal of ClosLjun, Janelia 646, and Deep Red was excited by a red laser ($\lambda_{ex}$ = 638 nm) and filtered through a 685/47 BPF before acquisition. Image J (https://imagej.net/ij/) was used to process microscopy images. The brightness and contrast setting were adjusted for the control images (Fig. S1 to S3) to maximize signal representation. In other words, the white point was minimized to capture any signal over background. For all other images, only the white point of the image was increased to minimize oversaturation in images of very bright samples. The black point and microscope power settings were never changed.

## ACKNOWLEDGMENTS

This work was supported by an ARPA-E project under contract AR0001505. J.D.H. was supported in part by a U.S. Department of Education GAANN Fellowship under grant P200A210065. Microscopy equipment was acquired with a shared instrumentation grant (grant S10 OD016361), and access was supported by the NIH-NIGMS (grant P20 GM103446), by the NSF (grant IIA-1301765), and by the state of Delaware. This research benefited from the BioStore data management resource at the University of Delaware Bioinformatics Data Science Core (RRID:SCR_017696) supported by an NIH shared instrumentation grant (NIH S10 OD028725) and Delaware INBRE (NIH P20 GM103446).

E.T.P. conceived the overall project. J.D.H. conceived the approach, worked with E.T.P for the experimental design, and carried out all experiments and data analysis. J.D.H. and E.T.P. interpreted the data and wrote the manuscript. J.D.H., Conceptualization, Investigation, Methodology, Formal analysis, Validation, Visualization, Resources, Data curation, Writing – original draft, review and editing | E.T.P., Conceptualization, Data curation, Formal analysis, Funding acquisition, Investigation, Methodology, Project administration, Resources, Supervision, Writing – review and editing.

## AUTHOR AFFILIATION

[1]Department of Chemical and Biomolecular Engineering, The Delaware Biotechnology Institute, University of Delaware, Newark, Delaware, USA

## AUTHOR ORCIDs

John D. Hill http://orcid.org/0000-0001-6127-3238

Eleftherios T. Papoutsakis http://orcid.org/0000-0002-1077-1277

## FUNDING

| Funder | Grant(s) | Author(s) |
|---|---|---|
| DOE | Advanced Research Projects Agency - Energy (ARPA-E) | AR0001505 | John D. Hill |
| | | Eleftherios T. Papoutsakis |
| U.S. Department of Education (ED) | P200A210065 | John D. Hill |
| HHS | NIH | National Institute of General Medical Sciences (NIGMS) | P20 GM103446 | John D. Hill |
| National Science Foundation (NSF) | IIA-1301765 | John D. Hill |
| HHS | National Institutes of Health (NIH) | S10 OD028725 | John D. Hill |
| Delaware IDeA Network of Biomedical Research Excellence (Delaware INBRE) | P20 GM103446 | John D. Hill |

## AUTHOR CONTRIBUTIONS

John D. Hill, Conceptualization, Data curation, Formal analysis, Investigation, Methodology, Resources, Validation, Visualization, Writing – original draft, Writing – review and editing | Eleftherios T. Papoutsakis, Conceptualization, Data curation, Formal analysis, Funding acquisition, Investigation, Methodology, Project administration, Resources, Supervision, Writing – review and editing

## ADDITIONAL FILES

The following material is available online.

### Supplemental Material

**Supplemental Figures (mSystems00572-24-s0001.docx).** Fig. S1 to S19.
**Supplemental Tables (mSystems00572-24-s0002.docx).** Tables S1 to S4.

### Open Peer Review

**PEER REVIEW HISTORY (review-history.pdf).** An accounting of the reviewer comments and feedback.

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
