## [Reviewer comments · mSystems]

Species-specific ribosomal RNA-FISH identifies interspecies cellular-material exchange, active-cell population dynamics and cellular localization of translation machinery in clostridial cultures and co-cultures

John Hill and Eleftherios Papoutsakis

Corresponding Author(s): Eleftherios Papoutsakis, University of Delaware

Review Timeline:

Submission Date:	April 19, 2024
Editorial Decision:	June 17, 2024
Revision Received:	August 1, 2024
Accepted:	August 7, 2024

Editor: Christopher Anderton

Reviewer(s): Disclosure of reviewer identity is with reference to reviewer comments included in decision letter(s). The following individuals involved in review of your submission have agreed to reveal their identity: Jan David Brüwer (Reviewer #1)

Transaction Report:

DOI: <https://doi.org/10.1128/msystems.00572-24>

Re: mSystems00572-24 (Species-Specific ribosomal RNA-FISH identifies interspecies cellular-material exchange, active-cell population dynamics and cellular localization of translation machinery in clostridial cultures and co-cultures)

Dear Prof. Eleftherios T Papoutsakis:

Overall, the Reviewers and myself found this manuscript to be very interesting and potentially an impactful paper. While my decision is "minor modifications", there are many inconsistencies in the writing, and some general editing due diligence is required for acceptance.

Revision Guidelines

Sincerely,
Christopher Anderton
Editor
mSystems

Reviewer #1 (Comments for the Author):

Hill and Papoutsakis used microscopy techniques to study the exchange of cytoplasmic content between different species in synthetic communities. I greatly appreciate their justifications throughout the manuscript and overall enjoyed reading the

manuscript. Their data is exiting and raises more questions than it answer (which should be understood as a compliment for good research).

I have a few questions and comments to the authors, which I hope will improve their study.

Title:
- "Species-specific". Any reasons why the 'specific' is capital?
- I would suggest to re-evaluate the title to focus on the ribosome and cytoplasmic material exchange. I think this is the core novelty in this manuscript, whereas 'cellular localisation of translation achinery' is less exciting.

Overall

- I think the underline for rRNA-FISH is not necessary.
- 'Clostridium' is the genus name, right? In that regard, it should always be capitalized and in italics throughout the manuscript. 'Clostridium spp.' refers to multiple species of the genus. 'Clostridia' is modern English and should not be italics. Please correct me if I am wrong here ...
- A recent publication in mSystems assesses cell division rates of microbes in the environment, using 16S rRNA FISH and a counterstain with a nucleic acid stain. They assess the ribosomal content of the microbes by assessing the fluorescence intensity of the cells and additionally determined the frequency of dividing cells. I think this research could be of great interest for the authors (elaborated multiple times below) and might spark new analysis ideas for future projects.
>> Brüwer, Jan D., Luis H. Orellana, Chandni Sidhu, Helena CL Klip, Cédric L. Meunier, Maarten Boersma, Karen H. Wiltshire, Rudolf Amann, and Bernhard M. Fuchs. "In situ cell division and mortality rates of SAR11, SAR86, Bacteroidetes, and Aurantivirga during phytoplankton blooms reveal differences in population controls." *Msystems* 8, no. 3 (2023): e01287-22.

Intro

- Refs 12 & 13 for I. 128: These references are very specific for flow cytometry and FISH. I think the text could use a more general reference for FISH (e.g., >>Glöckner, F. O., R. Amann, A. Alfreider, J. Pernthaler, R. Psenner, K. Trebesius, and K.-H. Schleifer. 1996. An in situ hybridization protocol for detection and identification of planktonic bacteria. *Systematic and Applied Microbiology* 19:403-406 or >>Fuchs, B. M., J. Pernthaler, and R. Amann. 2007. Single cell identification by fluorescence in situ hybridization, p. 886-896. In C. A. Reddy, T. J. Beveridge, J. A. Breznak, G. Marzluf, T. M. Schmidt, and L. R. Snyder (ed.), *Methods for General and Molecular Microbiology*, 3rd ed. ASM Press, Washington, D.C.)

Results

- I very much appreciated the extensive details and justifications regarding the method optimization and choices of methods. However, to help the reader understand, I would consider to move large parts of the results into the Materials and Methods (potentially even to Supplementary information for the dedicated reader). I see a greater value in the cell biology insights than the methodological advances ...
- On the same line, I am kind of missing a justification, why you chose HaloTag and CellTracker Deep Red, ...
- general: I think the hybrid cells are very interesting !! However, I am missing more negative controls, as the authors only observe 2 -3 % of the hybrid cells. 2 -3 % with very low FISH signal intensities could be an artefact, which really should be ruled out. I would suggest an experiment, where the authors take pure cultures, hybridize them with all three FISH probes simultaneously and report detected "hybrid" cells, too. I think this data could be hidden in Fig. 1 - but I am not too sure about this.

- I 172: Fuch>s< et al. The S is missing.

- I. 213 - 215 ("So, in essence, ..."). I am unsure what the sentence tries to communicate. Consider to rephrase it.
- II. 221 following. Just a comment: It would be possible to conduct FISH on polycarbonate filters and subsequently remove them. >>Sekar, Raju, Bernhard M. Fuchs, Rudolf Amann, and Jakob Pernthaler. "Flow sorting of marine bacterioplankton after fluorescence in situ hybridization." *Applied and Environmental Microbiology* 70, no. 10 (2004): 6210-6219.
- I. 235: I think it's unexpected that EtOH fixation and lysozyme digestion does not play well together. Could you elaborate on this? Do you have (unpublishable) data? I am curious what happened ...
- I. 289 "declined during at the onset". Please carefully read again ...
- I . 317 "In so doing" -> "In doing so"?
- I.326 (and others): I am curious about the unIID cells. Are ribosome numbers below the detection limit? Have the authors tried other FISH methods (e.g., more fluorophores per probe or CARD-FISH?) Also, I am curious about the proportion of identifiable cells in pure cultures of the strains. I guess Fig. 3 shows that after ~3 days, almost no cell could be detected with FISH anymore?
- I. 362: How do I know from Fig. 5c that the largest fraction was found after 4 h?
- I. 409: Almost all cells/chains contain the ClosKluy signal? Typo that "all but one" contain the ClosKluy probe?
- I. 424 following: I would suggest the authors to take a look into Brüwer et al., *mSystems* (as described above). The nucleoid could be visualized with a nucleic acid stain (DAPI, SYBR gold, ...), which could be interesting for future projects. Do the authors have any data for this already? I think this would be interesting ...
Using a nucleic acid stain, dividing cells could be visualized (which I acknowledge is beyond the scope of this paper!).
-I. 463: I don't see the information in Fig. S12?

Discussion

- The Discussion is very focused on the methods of 16S rRNA FISH and the probe design. I think this is to a large extent unnecessary here, as the authors did not develop a new method. I would rather suggest to focus on the molecular and cell biology insights. The exchange of cytoplasmic material - including ribosomes - is super interesting. I think the manuscript would benefit, if the discussion focuses on these processes. Is there literature or speculation on the exact mechanism?

What is the ecological benefit for the cells?

Why would they exchange ribosomes? This does not make sense to me ... Or is the exchange of ribosomes only a byproduct of exchange of other macromolecules?

As the authors focus on biotechnology in the introduction, interesting question could be about the usability of these processes?

What is the benefit for the industry? Can we regulate the process?

- I 537: You write it is important to conduct these experiments. But what is the conclusion from it?

Methods:

- please specify 'filter sterilized'. 0.2 µm?

- l. 616 space between 5> - In-solution RNA-FISH protocol: A reference to existing protocols is missing. In-solution FISH was not developed in this manuscript ..

- l. 681: Space between integers and units.

Figures

- general: I would suggest to use consistent colors for each stain throughout the manuscript. It's partly very confusing, what each color represents. In the end, all images are grayscale images with false colours - so it's interchangeable.

- general: I am curious why most images are so grainy and blurry. My impression is that especially the hybrid cells are heavily overexposed.

- Fig. 1

"labeled CAC"

D, E, F: What is the x axis?

- Fig. 4

Could the authors elaborate why Clostridium has a uniform distribution in panel F, whereas Clostridium does not? Why does the chain have so few Clostridium ribosomes, whereas the smaller cell is overexposed?

- Fig. 5:

C: What do the four different panels show?

E: Why is Clostridium in two different sub-images and why does it have such a different read-out in both?

- Fig 7:

The rRNA FISH signal is very distinct. Why do we not see such a pattern in any other images previously?

- Fig. 8:

Very interesting patterns, which are displayed!! Too bad, only three examples are provided. An automated images analysis of more cells would be cool to convince the reader of this pattern throughout the experimental set-up.

Supplementary figures: They are very dark and partly difficult to 'read'. Printing is almost impossible or useless ... Especially Fig. S1 - maybe it's possible to brighten the images up?

Instead of providing the full images - which I appreciate a lot! - as supplements, maybe it's worth to consider uploading them into an online repository?

Reviewer #2 (Comments for the Author):

This is an interesting paper that describes the development of a set of species-specific rRNA-FISH probes to follow the dynamics of artificial consortia of Clostridia species. Technically, it is excellent, demonstrating specificity and utility of their approach. However, it is poorly organized in places and presents results of variable quality. If it was streamlined and focused a bit better, it would make a strong contribution to the literature.

The authors start with the premise that species-specific probes are rarely being applied to industrially interesting Clostridia spp. (Clostridia species are also important in human health as well). The purpose of this study is to develop an rRNA-FISH protocol suitable for the routine analysis of the dynamics of synthetic consortia microbiology. They do an excellent job of laying out the problems and the steps necessary to make rRNA-FISH suitable for routine use. Particularly important was the deletion of unnecessary processing steps. They provide lots of good data in the supplements so that other scientists can determine if eliminating these steps will impact rRNA-FISH for other species. All their modifications make rRNA-FISH more practical. In general, their paper is an excellent general guide to designing and evaluating the use of rRNA-FISH probes for similar species.

After optimizing the rRNA-FISH protocol, they then explore its general usefulness. Here is where the logic of the paper becomes confusing. As a matter of course of validating their protocol, they make a series of observations or "discoveries" about the dynamics of rRNA in their cells, so they weave the discussion of the discoveries in with the validation of the technique, making the logic of the manuscript difficult to follow at times. For example, they needed to establish whether their rRNA-FISH probes only labeled live cells. The answer was yes and no. Their probes target the 23S rRNA, the primary scaffold rRNA of the large ribosomal subunit. Apparently, cells undergoing sporulation do not show evidence of labeling, suggesting that only cells able to proliferate can be labeled. They then go onto a discussion of the fraction of rRNA found in ribosomal precursor pools versus active ribosomes, versus inactive ribosomes to suggest that what their rRNA-FISH probes are detecting are centers of active protein synthesis. Then they go back to their validation studies, showing that the probes are compatible with protein labeling probes. This was very confusing and came across as describing the development of a tool to discover something they had already discovered. I suggest that they consolidate all the validation work in the first section before going on to the application and "discovery" part of the paper.

In general, the science in the "discovery" part of the paper is solid and they make some very interesting observations regarding the rRNA tags. First, they make a compelling case that there are indeed fusion events between species that cause the rRNA from one species to appear in another. The frequency is low, but both flow cytometry and microscopy provide compelling evidence. From the triple-mix experiment, the frequency seems to be proportional to species abundance, so it isn't clear whether this is a specific process or a random interchange. Clearly this is something that should be followed up with more targeted studies. The data on the dynamic localization/relocalization of the rRNA probes in the different species under different growth conditions are fascinating and is a highlight of the paper. It suggests a degree of localized protein synthesis in bacteria that is not generally appreciated. The studies that show that the rRNA-FISH probes only label actively growing cells are also interesting and convincing, although future studies should be conducted to understand the basis of this.

The experiments on using a combination of rRNA-FISH and protein (HaloTag) labeling to follow purported cytoplasmic fusions are far less convincing. The thresholds for flow cytometry are set differently for the proteins than for rRNA, complicating efforts to compare ratios. The puncta of HaloTag-OregonGreen looks more like a technical artifact, such as reagent precipitates, rather than transfer of proteins. The entire section on rRNA vs protein transfer (lines 336-385) is not very convincing and is extremely difficult to follow, especially because they are following different rRNA probes, protein probes and species that are being mixed and matched. They don't show any hybrids that display both rRNA and protein exchange, undercutting any broader interpretation of what they are seeing. Because cytoplasmic fusions are at best tangential to the rest of the paper, this section is best deleted.

Overall, this is a strong manuscript that needs some judicious editing. I do think it has the potential to make a significant contribution to the literature.

Minor points.

1. Need some grammatical editing as well. There are a number of extra or wrong words.
2. There is a meandering argument (lines 432-451) that compartmentalized rRNA-FISH fluorescence represents the compartmentalization of actively translating ribosomes. I get the gist of it, but it is difficult to follow. Suggest rewording.

This is an interesting paper that describes the development of a set of species-specific rRNA-FISH probes to follow the dynamics of artificial consortia of Clostridia species. Technically, it is excellent, demonstrating specificity and utility of their approach. However, it is poorly organized in places and presents results of variable quality. If it was streamlined and focused a bit better, it would make a strong contribution to the literature.

The authors start with the premise that species-specific probes are rarely being applied to industrially interesting Clostridia spp. (Clostridia species are also important in human health as well). The purpose of this study is to develop an rRNA-FISH protocol suitable for the routine analysis of the dynamics of synthetic consortia microbiology. They do an excellent job of laying out the problems and the steps necessary to make rRNA-FISH suitable for routine use. Particularly important was the deletion of unnecessary processing steps. They provide lots of good data in the supplements so that other scientists can determine if eliminating these steps will impact rRNA-FISH for other species. All their modifications make rRNA-FISH more practical. In general, their paper is an excellent general guide to designing and evaluating the use of rRNA-FISH probes for similar species.

After optimizing the rRNA-FISH protocol, they then explore its general usefulness. Here is where the logic of the paper becomes confusing. As a matter of course of validating their protocol, they make a series of observations or “discoveries” about the dynamics of rRNA in their cells, so they weave the discussion of the discoveries in with the validation of the technique, making the logic of the manuscript difficult to follow at times. For example, they needed to establish whether their rRNA-FISH probes only labeled live cells. The answer was yes and no. Their probes target the 23S rRNA, the primary scaffold rRNA of the large ribosomal subunit. Apparently, cells undergoing sporulation do not show evidence of labeling, suggesting that only cells able to proliferate can be labeled. They then go onto a discussion of the fraction of rRNA found in ribosomal precursor pools versus active ribosomes, versus inactive ribosomes to suggest that what their rRNA-FISH probes are detecting are centers of active protein synthesis. Then they go back to their validation studies, showing that the probes are compatible with protein labeling probes. This was very confusing and came across as describing the development of a tool to discover something they had already discovered. I suggest that they consolidate all the validation work in the first section before going on to the application and “discovery” part of the paper.

In general, the science in the “discovery” part of the paper is solid and they make some very interesting observations regarding the rRNA tags. First, they make a compelling case that there are indeed fusion events between species that cause the rRNA from one species to appear in another. The frequency is low, but both flow cytometry and microscopy provide compelling evidence. From the triple-mix experiment, the frequency seems to be proportional to species abundance, so it isn't clear whether this is a specific process or a random interchange. Clearly this is something that should be followed up with more targeted studies. The data on the dynamic localization/relocalization of the rRNA probes in the different species under different growth conditions are fascinating and is a highlight of the paper. It suggests a degree of localized protein synthesis in bacteria that is not generally appreciated. The studies that show that the rRNA-FISH probes only label actively

growing cells are also interesting and convincing, although future studies should be conducted to understand the basis of this.

The experiments on using a combination of rRNA-FISH and protein (HaloTag) labeling to follow purported cytoplasmic fusions are far less convincing. The thresholds for flow cytometry are set differently for the proteins than for rRNA, complicating efforts to compare ratios. The puncta of HaloTag-OregonGreen looks more like a technical artifact, such as reagent precipitates, rather than transfer of proteins. The entire section on rRNA vs protein transfer (lines 336-385) is not very convincing and is extremely difficult to follow, especially because they are following different rRNA probes, protein probes and species that are being mixed and matched. They don't show any hybrids that display both rRNA and protein exchange, undercutting any broader interpretation of what they are seeing. Because cytoplasmic fusions are at best tangential to the rest of the paper, this section is best deleted.

Overall, this is a strong manuscript that needs some judicious editing. I do think it has the potential to make a significant contribution to the literature.

Minor points.

1. Need some grammatical editing as well. There are a number of extra or wrong words.
2. There is a meandering argument (lines 432-451) that compartmentalized rRNA-FISH fluorescence represents the compartmentalization of actively translating ribosomes. I get the gist of it, but it is difficult to follow. Suggest rewording.

Reviewer #1 (Comments for the Author):

Hill and Papoutsakis used microscopy techniques to study the exchange of cytoplasmic content between different species in synthetic communities. I greatly appreciate their justifications throughout the manuscript and overall enjoyed reading the manuscript. Their data is exiting and raises more questions than it answer (which should be understood as a compliment for good research).

I have a few questions and comments to the authors, which I hope will improve their study.

Title:

- "Species-specific". Any reasons why the 'specific' is capital?

Response: Fixed. It was a typo. Thank you.

- I would suggest to re-evaluate the title to focus on the ribosome and cytoplasmic material exchange. I think this is the core novelty in this manuscript, whereas 'cellular localization of translation machinery' is less exciting.

Response: We agree that the cytoplasmic material exchange is the core novelty, but we think it's best to keep the title as is for a couple reasons. The other reviewer has expressed their opinion that the localization of translation machinery is a highlight of the work. Also, we feel that emphasizing the ability of the technique to track sub-populations will attract communities of interested researchers which would not have otherwise seen the work.

Overall

- I think the underline for rRNA-FISH is not necessary.

Response: Agreed. The underline has been removed.

- 'Clostridium' is the genus name, right? In that regard, it should always be capitalized and in italics throughout the manuscript. 'Clostridium spp.' refers to multiple species of the genus. 'Clostridia' is modern English and should not be italics. Please correct me if I am wrong here ...

Response: Correct. We should be capitalizing "Clostridia(!)" and more careful about using "Clostridium spp." Corrections made on lines 53, 77, 78, 86, 154, 229, 258, 493, 501, 503, 549, 554 (of the original manuscript). Thank you!

- A recent publication in mSystems assesses cell division rates of microbes in the environment, using 16S rRNA FISH and a counterstain with a nucleic acid stain. They assess the ribosomal content of the microbes by assessing the fluorescence intensity of the cells and additionally determined the frequency of dividing cells. I think this research could be of great interest for the authors (elaborated multiple times below) and might spark new analysis ideas for future projects.

>> Brüwer, Jan D., Luis H. Orellana, Chandni Sidhu, Helena CL Klip, Cédric L. Meunier, Maarten Boersma, Karen H. Wiltshire, Rudolf Amann, and Bernhard M. Fuchs. "In situ cell division and mortality rates of SAR11, SAR86, Bacteroidetes, and Aurantivirga during phytoplankton blooms reveal differences in population controls." *Msystems* 8, no. 3 (2023): e01287-22.

Response: Thank you for bringing this interesting paper to our attention. From the way we read the paper, their objective is to report the two attributes of net growth rate: actual growth rate and mortality (death) rate. These provide insight into the population control mechanisms during phytoplankton blooms in the German archipelago. This approach would be very useful for future experiments as the metabolic regulation changes through the life cycle of Clostridium acetobutylicum, whether in monoculture or cocultures. We have included some discussion of this approach in the discussion.

Intro

- Refs 12 & 13 for I. 128: These references are very specific for flow cytometry and FISH. I think the text could use a more general reference for FISH (e.g., >>Glöckner, F. O., R. Amann, A. Alfreider, J. Pernthaler, R. Psenner, K. Trebesius, and K.-H. Schleifer. 1996. An in situ hybridization protocol for detection and identification of planktonic bacteria. *Systematic and Applied Microbiology* 19:403-406 or

>>Fuchs, B. M., J. Pernthaler, and R. Amann. 2007. Single cell identification by fluorescence in situ hybridization, p. 886-896. In C. A. Reddy, T. J. Beveridge, J. A. Breznak, G. Marzluf, T. M. Schmidt, and L. R. Snyder (ed.), Methods for General and Molecular Microbiology, 3rd ed. ASM Press, Washington, D.C.)

Response: Agreed. We had hoped to give due credit to this group of authors (Fuchs, Glockner, Amann, Wallner, etc.) for their contributions to FISH methods which greatly supported our work. We have now done so.

Results

- I very much appreciated the extensive details and justifications regarding the method optimization and choices of methods. However, to help the reader understand, I would consider to move large parts of the results into the Materials and Methods (potentially even to Supplementary information for the dedicated reader). I see a greater value in the cell biology insights than the methodological advances ...

Response: Thank you for the suggestion. We have done so to some extent.

- On the same line, I am kind of missing a justification, why you chose HaloTag and CellTracker Deep Red, ...

Response: Yes, we have not adequately justified this decision and have modified the manuscript to correct that. Briefly, GFP, mCherry, and the flavin-binding proteins have been used with limited success in anaerobic culture, so our lab developed the HaloTag reporter systems for Clostridium acetobutylicum and Clostridium ljungdahlii in 2020. We have used HaloTag and CellTracker DeepRed subsequently to visualize the heterologous cell fusion behavior and use them again in this study. So, they were the most appropriate choices.

- general: I think the hybrid cells are very interesting !! However, I am missing more negative controls, as the authors only observe 2 -3 % of the hybrid cells. 2 -3 % with very low FISH signal intensities could be an artefact, which really should be ruled out. I would suggest an experiment, where the authors take pure cultures, hybridize them with all three FISH probes simultaneously and report detected "hybrid" cells, too. I think this data could be hidden in Fig. 1 - but I am not too sure about this.

Response: It was important to us to avoid reporting false positives. The experiment which you recommend is indeed the experiment described in Fig. 1D,E,F, though we presented the results visually as flow cytometric histograms. We have added a supplemental Tables S1 and S2 to quantitatively demonstrate the absence of false "hybrid" cells in the pure cultures. The maximum false positive rates for any species/probe combination was 0.26%, but most others were below 0.1%.

-l 172: Fuch>s< et al. The S is missing.

Response: Corrected.

- l. 213 - 215 ("So, in essence, ..."). I am unsure what the sentence tries to communicate. Consider to rephrase it.

Response: We took care of that. Thank you!

- ll. 221 following. Just a comment: It would be possible to conduct FISH on polycarbonate filters and subsequently remove them. >>Sekar, Raju, Bernhard M. Fuchs, Rudolf Amann, and Jakob Pernthaler. "Flow sorting of marine bacterioplankton after fluorescence in situ hybridization." Applied and Environmental Microbiology 70, no. 10 (2004): 6210-6219.

Response: Thank you for pointing this out!

- l. 235: I think it's unexpected that EtOH fixation and lysozyme digestion does not play well together. Could you elaborate on this? Do you have (unpublishable) data? I am curious what happened ...

Response: In the early stages of the project, we tested the effect of lysozyme treatment duration (0, 10, 30, 60, 120 minutes) on ethanol fixed, PFA fixed, and unfixed samples following the protocol of Schneider et al. (2021) (Ref. 18). We found that the lysozyme treatment had little effect on the brightness of the PFA fixed and unfixed samples at all treatment durations. In the ethanol fixed cells, the untreated cells were bright. However, all lysozyme treated cells, even for 10 mins, exhibited very little fluorescence. We chose not to publish these data since it was captured with a ThermoFisher EVOS cell imaging system, which, while very simple/fast to use, produces poor images of microbial cells due to their small size. Moreover, Schneider et al. (2021) (Ref. 18) had already reported that prolonged lysozyme treatment was detrimental to ethanol fixed samples in Clostridium carboxidivorans and Clostridium kluyveri, so we did not feel it was a substantial contribution. We have included the images here:

- I. 289 "declined during at the onset". Please carefully read again ...

Response: Corrected. This should read, "declined at the onset."

- I. 317 "In so doing" -> "In doing so"?

Response: Technically, they have identical meanings. We prefer "in so doing," to emphasize the modification described in the preceding sentence. "In doing so" emphasizes the subsequent clause, "C. kluyveri becomes dependent..." in our opinion.

- I.326 (and others): I am curious about the unID cells. Are ribosome numbers below the detection limit? Have the authors tried other FISH methods (e.g., more fluorophores per probe or CARD-FISH?) Also, I am curious about the proportion of identifiable cells in pure cultures of the strains. I guess Fig. 3 shows that after ~3 days, almost no cell could be detected with FISH anymore?

Response: Yes, the ribosomes numbers are below detection limits. We did not try other FISH techniques like CARD-FISH but I am not sure how the increase in sensitivity would have resulted in significantly different results. Possibly a fraction of unID cells are false-negatives, but that is probably a very low number because the fluorescent labelling in monoculture shows that FISH can identify cells past the point of metabolic activity in batch cultures.

- I. 362: How do I know from Fig. 5c that the largest fraction was found after 4 h?

Response: Apologies. Supplemental Fig. S7 has the 11 hour time point which has a lower fraction. It should have been referenced as well. We have moved the original Fig 5 to the supplemental materials in an effort to streamline the manuscript.

- I. 409: Almost all cells/chains contain the ClosKluy signal? Typo that "all but one" contain the ClosKluy probe?

Response: We report that almost all cells contain the ClosLjun probe, but only some cells contain the ClosKluy probe. We modified the manuscript to clarify.

- I. 424 following: I would suggest the authors to take a look into Brüwer et al., mSystems (as described above). The nucleoid could be visualized with a nucleic acid stain (DAPI, SYBR gold, ...), which could be interesting for future projects. Do the authors have any data for this already? I think this would be interesting ...

Using a nucleic acid stain, dividing cells could be visualized (which I acknowledge is beyond the scope of this paper!).

Response: Thank you for the recommendation! We have considered using DAPI, but do not done so. We plan to use DAPI in the future.

-I. 463: I don't see the information in Fig. S12?

Response: Figure S12 is now Fig. S13. The microscopic images presented in Fig. S13 are meant to corroborate the drop in fluorescent signal observed by flow cytometry. Fluorescent signal could not be observed in the vast majority of cells which is why the microscopy images in Fig. S13 appear as black squares.

Discussion

- The Discussion is very focused on the methods of 16S rRNA FISH and the probe design. I think this is to a large extend unnecessary here, as the authors did not develop a new method.

Response: The purpose of discussing the methodology of rRNA-FISH is to bring attention to the tool's usefulness and practicality. To microbiologists in the fields of ecology and pathology, this is probably obvious, but rRNA-FISH has not been properly appreciated in the field industrial microbiology and biotechnology, an audience this paper is aiming at.

I would rather suggest to focus on the molecular and cell biology insights. The exchange of cytoplasmic material - including ribosomes - is super interesting. I think the manuscript would benefit, if the discussion focuses on these processes. Is there literature or speculation on the exact mechanism?

Response: Thank you. We are fascinated by the behavior as well. At this time, we are the only group who has reported this phenomenon and we have not been able to elucidate the mechanism. Some similar work has been published from Marie Therese Giudici-Ortoni's lab at Aix-Marseille University which implicates quorum sensing molecules. We discussed this work in the introduction, but there are key differences (described in the introduction) between their findings and ours that make us skeptical of any conclusions that we may be drawn by comparisons between them.

What is the ecological benefit for the cells?

Response: Thank you for this question. We are interested in answering it as well. We agree that there must be an ecological reason for that. Our basic hypothesis is that cytoplasmic exchange removes the cell membrane barrier in the anaerobic food chain: ethanol and acetate from C. ljungdahlii being accessed by C. kluveri without transport barriers, that is without first being released into the medium and then taken up by C. kluveri. This may be driven by chemotactic attraction similar to what we have suggested, based on some evidence, for the C. ljungdahlii-C. acetobutylicum pair in our 2024 mBio paper (ref. 10 of the original submission). That is, C. kluveri is attracted to the source of ethanol and acetate, which is C. ljungdahlii. This may provide an ecological benefit to the two organisms involved: C. ljungdahlii grows by forming acetate and ethanol, which would inhibit cell growth if they accumulate to high levels. C. kluveri grows on acetate and ethanol and prevents them from accumulating and thus becoming inhibitory. C. kluveri produces largely hexanoate, by removing 2 mol of ethanol and 1 mol of acetate per mol hexanoate, which while also inhibitory cannot accumulate to very high levels due to its low solubility. All that of course remains to be validated.

Why would they exchange ribosomes? This does not make sense to me ... Or is the exchange of ribosomes only a byproduct of exchange of other macromolecules?

Response: With the limited evidence on hand, we think that trying to address these questions would be too speculative. As suggested, the simplest explanation might be that it is a byproduct of other macromolecular exchange. One could speculate that it is possible that hybrid ribosomes are formed, and thus cells may use this as a mechanism to optimize ribosomal function.

As the authors focus on biotechnology in the introduction, interesting question could be about the usability of these processes? What is the benefit for the industry? Can we regulate the process?

Response: Use of this approach is most desirable for tracking population dynamics and stability in industrial or large-scale applications of both mono-cultures and mixed cultures. For monocultures, the interest is to understand culture productivity in stationary phase of sporulating and non-sporulating organisms especially for organisms (like clostridia) for which we do not have reliable or efficient plating assays. Also, there are several processes in environmental biotechnology using mixed cultures (eg, DOI 10.3389/fmicb.2012.00203, DOI 10.1016/j.biortech.2021.124985), and several using synthetic mixed cultures (eg., DOI 10.1186/s12934-019-1083-3). We have added to the discussion to this effect.

- I 537: You write it is important to conduct these experiments. But what is the conclusion from it?

Response: Perhaps we should have use the term "important" rather than "essential". We do now. Understanding the ribosomal localization and its dynamics will help us interpret ribosomal localization and dynamics in mixed cultures that result in exchange of cytoplasmic material including ribosomes; and the potential formation of hybrid ribosomes. Differences in ribosome localization between different species of the genus is also interesting from the physiological point of view. Are there genomic and/or physiological traits that could account for such differences? Also, one would like to examine the stability of hybrid cells and ribosomes and explore it as a potential mechanism of evolutionary biology. Understanding these mechanisms might allow us to use them synthetically to create cells with designed ribosomes to achieve synthetic goals. If at all possible, an interesting possibility is to examine the localization of ribosomes as the two cell types come to close proximity prior to heterologous fusion.

Methods:

- please specify 'filter sterilized'. 0.2 µm?

Response: Corrected

- l. 616 space between 5> <M HCl. (Space missing)

Response: Corrected

- In-solution RNA-FISH protocol: A reference to existing protocols is missing. In-solution FISH was not developed in this manuscript ..

Response: Corrected

- l. 681: Space between integers and units.

Response: Corrected

Figures

-general: I would suggest to use consistent colors for each stain throughout the manuscript. It's partly very confusing, what each color represents. In the end, all images are grayscale images with false colours - so it's interchangeable.

Response: Consistent colors were used throughout the manuscript. ClosLjun - RED. ClosKluy - GREEN. ClosAcet - MAGENTA. Janelia 549 - YELLOW. Janelia 646 - RED. DeepRed - RED. OregonGreen - GREEN.

- general: I am curious why most images are so grainy and blurry. My impression is that especially the hybrid cells are heavily overexposed.

Response: The size of the cells and the required resolution to capture them makes each pixel very dim.

- Fig. 1
"labeled CAC"

Response: Corrected

D, E, F: What is the x axis?

Response: Fluorescent intensity. Figure capture was modified to explain this.

- Fig. 4
Could the authors elaborate why ClosKluy has a uniform distribution in panel F, whereas ClosLjun does not? Why does the chain have so few ClosLjun ribosomes, whereas the smaller cell is overexposed?

Response: Corrected. We elaborate on possible causes of this in the results section.

- Fig. 5:
C: What do the four different panels show?
E: Why is ClosAcet in two different sub-images and why does it have such a different read-out in both?

Response: Corrected. This was a mistake in the image's labelling. Thank you for pointing this out. It should read "ClosLjun" in the left-most sub-images.

- Fig 7:
The rRNA FISH signal is very distinct. Why do we not see such a pattern in any other images previously?
Response: The pattern appears in Fig. 4E. However, it is possible that the formation of "hybrid cells" disrupts subcellular localization.

- Fig. 8:
Very interesting patterns, which are displayed!! Too bad, only three examples are provided. An automated images analysis of more cells would be cool to convince the reader of this pattern throughout the experimental set-up.

Response: Thank you! We have included the full frame images in the supplemental figures which provide many more examples to support our findings.

Supplementary figures: They are very dark and partly difficult to 'read'. Printing is almost impossible or useless ... Especially Fig. S1 - maybe it's possible to brighten the images up?
Instead of providing the full images - which I appreciate a lot! - as supplements, maybe it's worth to consider uploading them into an online repository?

Response: The repository is a great idea we had not considered. We now provide links to a repository for the supplemental Fig. S1, S2, and S3.

Reviewer #2 (Comments for the Author):

This is an interesting paper that describes the development of a set of species-specific rRNA-FISH probes to follow the dynamics of artificial consortia of Clostridia species. Technically, it is excellent, demonstrating specificity and utility of their approach. However, it is poorly organized in places and presents results of variable quality. If it was streamlined and focused a bit better, it would make a strong contribution to the literature.

Response: Thank you very much!

The authors start with the premise that species-specific probes are rarely being applied to industrially interesting Clostridia spp. (Clostridia species are also important in human health as well). The purpose of this study is to develop an rRNA-FISH protocol suitable for the routine analysis of the dynamics of synthetic consortia microbiology. They do an excellent job of laying out the problems and the steps necessary to make rRNA-FISH suitable for routine use. Particularly important was the deletion of unnecessary processing steps. They provide lots of good data in the supplements so that other scientists can determine if eliminating these steps will impact rRNA-FISH for other species. All their modifications make rRNA-FISH more practical. In general, their paper is an excellent general guide to designing and evaluating the use of rRNA-FISH probes for similar species.

Response: Again, thank you.

After optimizing the rRNA-FISH protocol, they then explore its general usefulness. Here is where the logic of the paper becomes confusing. As a matter of course of validating their protocol, they make a series of observations or "discoveries" about the dynamics of rRNA in their cells, so they weave the discussion of the discoveries in with the validation of the technique, making the logic of the manuscript difficult to follow at times. For example, they needed to establish whether their rRNA-FISH probes only labeled live cells. The answer was yes and no. Their probes target the 23S rRNA, the primary scaffold rRNA of the large ribosomal subunit. Apparently, cells undergoing sporulation do not show evidence of labeling, suggesting that only cells able to proliferate can be labeled. They then go onto a discussion of the fraction of rRNA found in ribosomal precursor pools versus active ribosomes, versus inactive ribosomes to suggest that what their rRNA-FISH probes are detecting are centers of active protein synthesis. Then they go back to their validation studies, showing that the probes are compatible with protein labeling probes. This was very confusing and came across as describing the development of a tool to discover something they had already discovered.

I suggest that they consolidate all the validation work in the first section before going on to the application and "discovery" part of the paper.

Response: We agree with the suggestion that a clear delineation between validation and discovery would lead to less confusion and that a consolidation of the "validation" portion into a single section is warranted. We have moved relevant portions of the validation section to materials and methods and the discussion to streamline the text. In the results section, the compatibility with protein labelling probes is given first, then we describe the evidence for the correlation between labelling and viability.

In general, the science in the "discovery" part of the paper is solid and they make some very interesting observations regarding the rRNA tags. First, they make a compelling case that there are indeed fusion events between species that cause the rRNA from one species to appear in another. The frequency is low, but both flow cytometry and microscopy provide compelling evidence. From the triple-mix experiment, the frequency seems to be proportional to species abundance, so it isn't clear whether this is a specific process or a random interchange. Clearly this is something that should be followed up with more targeted studies.

Response: We agree that this would be an interesting direction to pursue in the future.

The data on the dynamic localization/relocalization of the rRNA probes in the different species under different growth conditions are fascinating and is a highlight of the paper. It suggests a degree of localized protein synthesis in bacteria that is not generally appreciated. The studies that show that the rRNA-FISH probes only label actively growing cells are also interesting and convincing, although future studies should be conducted to understand the basis of this.

Response: Thank you! We agree that a more pointed inquiry would likely yield interesting insights.

The experiments on using a combination of rRNA-FISH and protein (HaloTag) labeling to follow purported cytoplasmic fusions are far less convincing. The thresholds for flow cytometry are set differently for the proteins than for rRNA, complicating efforts to compare ratios.

Responses: The thresholds are set according to the fluorescence intensity typically observed by a labelled cell using the probe of interest. The thresholds used for the rRNA-FISH probes were set based on the negative and positive controls as illustrated in Fig. 1. The thresholds for the OregonGreen HaloTag Complex were set by positive and negative controls in Supplemental Figure S6. The thresholds for Oregon green are much lower than for ClosLjun because C. acetobutylicum exhibited much less green background fluorescence than red background fluorescence.

The puncta of HaloTag-OregonGreen looks more like a technical artifact, such as reagent precipitates, rather than transfer of proteins.

Response: This is a valuable insight. Thank you. We had not considered the possibility of reagent precipitates largely because the microscopic images of the in Supplemental figure S1, S2, and S3 did not have any random puncta. We discuss this possibility in the relevant section of the supplementary results.

The entire section on rRNA vs protein transfer (lines 336-385) is not very convincing and is extremely difficult to follow, especially because they are following different rRNA probes, protein probes and species that are being mixed and matched. They don't show any hybrids that display both rRNA and protein exchange, undercutting any broader interpretation of what they are seeing. Because cytoplasmic fusions are at best tangential to the rest of the paper, this section is best deleted.

Response: Our primary motivation to report these results is to address the common argument that flow cytometry may be overestimating the frequency of hybrid cells since cell agglomerates would be positive for both fluorescent markers. This experiment shows that at least the vast majority of hybrid cells cannot be attributed to cell agglomerates. With that being said, we agree with your suggestion that the section may be hard to follow. Therefore we have moved this section to the supplemental material.

Overall, this is a strong manuscript that needs some judicious editing. I do think it has the potential to make a significant contribution to the literature.

Response: Thank you very much!

Minor points.

1. Need some grammatical editing as well. There are a number of extra or wrong words.

Response: Thank you, we have corrected these.

2. There is a meandering argument (lines 432-451) that compartmentalized rRNA-FISH fluorescence represents the compartmentalization of actively translating ribosomes. I get the gist of it, but it is difficult to follow. Suggest rewording.

Response: Thank you! We have reworded it to better clarify the message.

Re: mSystems00572-24R1 (Species-specific ribosomal RNA-FISH identifies interspecies cellular-material exchange, active-cell population dynamics and cellular localization of translation machinery in clostridial cultures and co-cultures)

Dear Prof. Eleftherios T Papoutsakis:

Your manuscript has been accepted, and I am forwarding it to the ASM production staff for publication. Your paper will first be checked to make sure all elements meet the technical requirements. ASM staff will contact you if anything needs to be revised before copyediting and production can begin. Otherwise, you will be notified when your proofs are ready to be viewed.

Sincerely,
Christopher Anderton
Editor
mSystems